# Single cell transcriptomic analysis of HPV16-infected epithelium identifies a keratinocyte subpopulation implicated in cancer

Mary C. Bedard[1,27], Tafadzwa Chihanga[1,27], Adrean Carlile[1], Robert Jackson [2], Marion G. Brusadelli[3], Denis Lee[4], Andrew VonHandorf[5], Mark Rochman [6], Phillip J. Dexheimer [7], Jeffrey Chalmers [8], Gerard Nuovo[9], Maria Lehn [10], David E. J. Williams[11,12], Aditi Kulkarni[13,14], Molly Carey[15], Amanda Jackson[15], Caroline Billingsley[15], Alice Tang[16], Chad Zender[16], Yash Patil[16], Trisha M. Wise-Draper[10], Thomas J. Herzog[15], Robert L. Ferris[13,14,17,18], Ady Kendler[19], Bruce J. Aronow [7,20,21], Matthew Kofron[20,21], Marc E. Rothenberg [6,21], Matthew T. Weirauch[5,21,22], Koenraad Van Doorslaer [2,11,23,24,25], Kathryn A. Wikenheiser-Brokamp [19,26], Paul F. Lambert[4], Mike Adam [20] ✉, S. Steven Potter [20] ✉ & Susanne I. Wells [1,21] ✉

Persistent HPV16 infection is a major cause of the global cancer burden. The viral life cycle is dependent on the differentiation program of stratified squamous epithelium, but the landscape of keratinocyte subpopulations which support distinct phases of the viral life cycle has yet to be elucidated. Here, single cell RNA sequencing of HPV16 infected compared to uninfected organoids identifies twelve distinct keratinocyte populations, with a subset mapped to reconstruct their respective 3D geography in stratified squamous epithelium. Instead of conventional terminally differentiated cells, an HPV-reprogrammed keratinocyte subpopulation (HIDDEN cells) forms the surface compartment and requires overexpression of the ELF3/ESE-1 transcription factor. HIDDEN cells are detected throughout stages of human carcinogenesis including primary human cervical intraepithelial neoplasias and HPV positive head and neck cancers, and a possible role in promoting viral carcinogenesis is supported by TCGA analyses. Single cell transcriptome information on HPV-infected versus uninfected epithelium will enable broader studies of the role of individual keratinocyte subpopulations in tumor virus infection and cancer evolution.

Infection with high-risk human papillomavirus (HPV) types causes 5% of all cancers worldwide, including cervical and oropharyngeal cancers[1]. HPV16 accounts for the majority, with persistent keratinocyte infection being a primary risk factor[2–6]. The role of the viral oncogenes in human carcinogenesis[7–12] has been intensely investigated for decades, and the importance of differentiation for the viral life cycle is known[13–18]. However, the transcriptomic landscape of heterogeneous stratified squamous epithelial subpopulations which support different phases of the viral life cycle has yet to be described, as bulk sequencing approaches[19,20] preclude a granular understanding of the distinct keratinocyte cell types comprising the host epithelium.

In this work, we report the transcriptomic landscape of isogenic HPV16-infected versus uninfected 3D stratified squamous epithelium[21] and demonstrate its utility in discovering a keratinocyte subpopulation that is greatly amplified and reprogrammed by HPV, termed HPV-induced differentiation-dissonant epithelial nonconventional (HIDDEN) cells. This subpopulation forms an unexpected distinct superficial cellular compartment in HPV-infected epithelial models; the same compartment is not detected in their uninfected counterparts or in primary human oropharyngeal or cervical mucosa. HIDDEN cells persist through multiple stages of human carcinogenesis, including p16+ cervical intraepithelial neoplasia (CIN) lesions and HPV+ head and neck squamous cell cancers (HNSCC) (visual summary, Fig. 1a). TCGA analyses identify a signature of co-occurring biomarkers shared between HPV-infected and HPV-transformed tissues, and correlate high expression of these biomarkers with worse clinical outcomes. Using the HIDDEN cell transcriptome to identify candidate master regulators, we demonstrate that depletion of ELF3/ESE-1 in HPV+ epithelium greatly reduces HIDDEN cell biomarker expression and compartment formation. Altogether, our studies support an important role for this keratinocyte compartment in HPV-driven disease and for ELF3 as a potential therapeutic target in the fight against HPV persistence. The single-cell transcriptome data will serve as a key resource in the field of oncogenic viruses, wherein keratinocyte subpopulations can now be individually investigated for their roles in supporting viral life cycles and multistep cancer evolution.

## Results

### scRNAseq reveals enrichment of distinct keratinocyte subpopulations in HPV16+ versus HPV16− stratified squamous epithelial rafts

For single-cell RNA sequencing (scRNAseq) experiments, we selected a well-established 3D organotypic epithelial raft model which recapitulates the stratified squamous epithelium required for differentiation-dependent HPV viral gene expression and replication (Fig. 1b). Normal immortalized keratinocytes (NIKS)[22] and derivative, isogenic HPV16+ NIKS were used to generate HPV16− and HPV16+ epithelial rafts, respectively (Fig. 1c). Histopathological features of HPV16 infection were confirmed to be present by a board-certified pathologist in HPV16+ but not HPV16− rafts ($n = 4$ independent sets of rafts), including atypical epithelial cells with koilocytic features and mitotic figures within suprabasal epithelial cells. Genomic HPV DNA was uniformly present in HPV16+ but not HPV16− rafts by in situ hybridization (DNA-ISH, Fig. 1d).

Very few cells stained positive for viral episomes in terminally differentiated layers, demonstrating the rare occcurrence of the productive phase of the HPV viral life cycle[22]. This was further supported by RT-qPCR analysis, where early viral E1, E6, and E7 gene expression was more robust than late viral L1 gene expression (Fig. 1e), and by Southern blot analysis, which showed a modest increase in episomal HPV DNA in 3D rafts when compared to monolayer 2D cultures (Fig. 1f). Taken together, these data demonstrate persistent maintenance of infection in HPV+ rafts with rare cells supporting productive viral replication. Additionally, HPV16 viral activities were evident by increased expression of two established clinical markers of HPV+ tissues, exportin 5[23] (Fig. 1g) and p16[24] (Fig. 1h), known to be induced by HPV infection.

One matched pair of isogenic HPV16+ and HPV16− rafts was harvested on day 14 after keratinocytes were plated, separately dissociated into single-cell suspensions, and subjected independently to 10x Genomics Chromium RNA sequencing (Fig. 2a). Five additional sets of matched rafts were dissociated into single-cell suspensions and snap-frozen for validation studies. Following batch correction, the combined UMAP showed a high degree of overlap between HPV16+ and HPV16− cells (Fig. 2b), reflecting the known maintenance of the overall stratified squamous epithelial morphology upon HPV infection.

HPV16 early (E1, E2, E5, E6, and E7) and late (L1 and L2) gene expression was confined to cells originating from HPV16+ rafts with a preponderance of cells expressing early viral genes (Fig. 2c), in alignment with previous RT-qPCR results (Fig. 1e). Further analysis identified twelve transcriptomically distinct clusters of keratinocyte subpopulations that were designated clusters 0–11 (C0–C11), with most clusters containing cells originating from both HPV16− and HPV16+ rafts (Fig. 2d and Supplementary Data 2). Interestingly, C9 was uniquely separated from the remaining 11 clusters (Fig. 2e) and was highly enriched in HPV16+ cells (72% of C9 cells, Fig. 2f). In contrast, four clusters (C0–C3) had relatively similar numbers of HPV16+ versus HPV16− cells, three clusters (C4, C5, and C7) were enriched in HPV16− cells, and two clusters (C10 and C11) had too few cells to characterize with certainty. In addition to C9, subpopulation C6 was also enriched in HPV16+ rafts (67% of C6 cells) and subpopulation C8 was exclusive to HPV16+ rafts (>95% of C8 cells).

### Keratinocyte subpopulations are primarily defined by the differentiation program of stratified squamous epithelium

Pseudotemporal trajectory analysis orders cells along a spectrum describing progression through a biological process[25]. All clustered keratinocyte subpopulations were found to align along a single continuum of the squamous epithelial differentiation program (Fig. 3a), with the exception of C9. Although C9 was initially included in pseudotime analysis (trajectory Supplementary Fig. 1a, heatmap Supplementary Fig. 1b, cluster plot Supplementary Fig. 1c, and gene expression Supplementary Fig. 1d), its drastically divergent transcriptome results in non-contiguous positioning in the UMAP and thus it could not be included in robust pseudotime analyses. The exclusion from the conventional epithelial differentiation program represented by the pseudotime trajectory resulted in C9 cells being termed differentiation dissonant. This description was reinforced by low C9 expression of conventional markers of the layers of the epithelium (Supplementary Fig. 1d).

In the final pseudotime analysis, gene expression signatures along the pseudotime trajectory progressed from basal (CO17A1+ and ITGB4+) to suprabasal (CARD18+ and KLK7+) cell states as evidenced by characteristic markers established in the literature (Fig. 3b). When clusters were plotted by pseudotime value (Fig. 3c), they grouped into populations with either more basal cell (C10, C7, C6, C4, and C5) or differentiated cell characteristics (C1, C8, C0, C11, C2, and C3).

Layer-specific epidermal markers routinely used at the protein level, namely COL17A and ITGB4 for the stratum basale, SBSN and DMKN for the stratum spinosum, CARD18 and DSG1 for the stratum granulosum, and KLK5 and KLK7 for the stratum corneum, were queried to define the cell clusters based upon RNA expression. Expression levels across pseudotime (Fig. 3d and Supplementary Fig. 1e) and in the UMAP (Fig. 3e and Supplementary Fig. 1f) confirmed that these known protein-level layer markers were captured in the scRNAseq analysis and that mRNA expression across rafts mirrored their protein-level progression from basal to terminally differentiated cell states. Proper laminar differentiation and recapitulation of stratified squamous layers in rafts was examined by IHC staining for protein expression of layer-specific markers in HPV16− and HPV16+ rafts (Fig. 3f). The protein staining pattern in HPV16− rafts mirrored normal human epidermis, demonstrating that 3D rafts recapitulate the cellular processes and architecture of the native in vivo tissue (Fig. 3g). Laminar cellular differentiation was also present in the HPV16+ rafts by protein expression analysis, consistent with the retained epithelial stratification demonstrated by histological analysis (Fig. 1c)[22,26]. Of note, however, HPV16+ epithelium had less intense staining of differentiation markers (CARD18 and KLK5), demonstrating that HPV reprogramming suppresses the normal differentiation program[26–29], particularly in the uppermost layers of the epithelium.

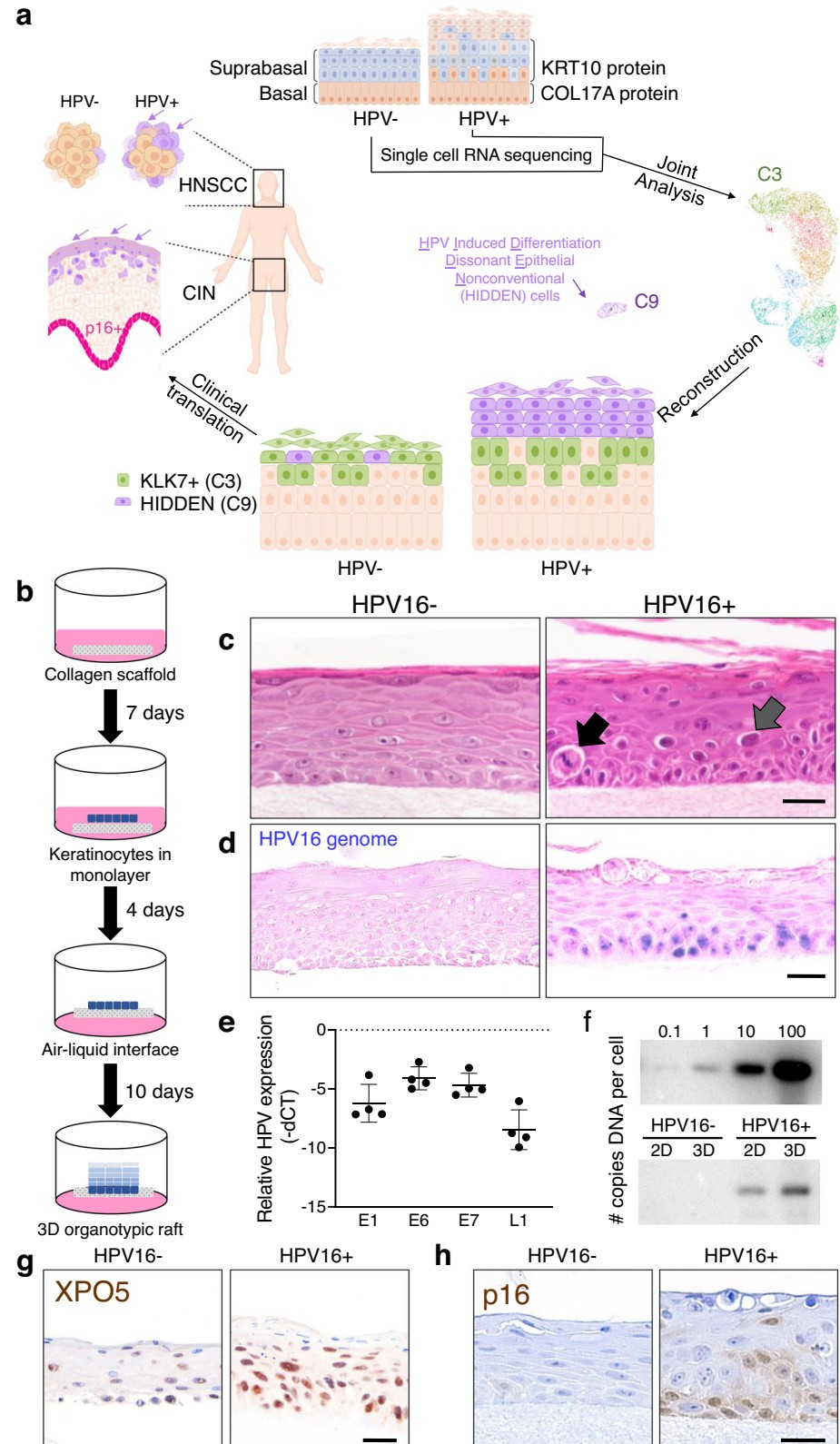

## Spatial mapping of suprabasal cell clusters reveals a yet undescribed epithelial compartment in HPV16+ rafts

By definition, scRNAseq data does not provide geographical localization of subpopulations in 3D tissues. We used RNA-ISH to spatially locate select keratinocyte subpopulations identified by scRNAseq within the 3D geography of HPV16+ and HPV16− rafts. Given the differential protein staining observed in suprabasal HPV+ vs HPV− rafts

(Fig. 3f), the uppermost differentiated subpopulations were of particular interest. We hypothesized that differences in geographical distribution could be reflective of transcriptional reprogramming by HPV and also underlie protein-level changes observed in Fig. 3f. To this end, C3 was identified as the most terminally differentiated cell subpopulation (Fig. 4a) and selected for validation and spatial localization. While each scRNAseq cluster is defined cumulatively by a distinct

**Fig. 1 | Generation of isogenic HPV16+ versus HPV16- squamous epithelia.**
**a** Visual summary. Created with BioRender.com. **b** Schematic of 3D organotypic epithelial raft generation using HPV16+ and HPV16− NIKS on a collagen scaffold with embedded fibroblasts. **c** Representative H&E stained images from $n = 4$ raft sets showing that HPV16+ rafts harbor atypical epithelial cells with koilocytic features, including cellular enlargement, hyperchromatic nuclei, and perinuclear clearing (gray arrow) as well as mitotic cells within the suprabasal layers (black arrow) that are not seen in HPV16− rafts. **d** Representative DNA-ISH images for HPV16 genome sequences showing HPV detection in rafts ($n = 2$ independent raft sets) generated from HPV16+, but not HPV16−, keratinocytes. **e** RT-qPCR ($n = 4$

HPV16+ rafts) for viral genes showing strong expression of early viral genes E1, E6, and E7 when compared to the late viral L1 gene, in line with a largely non-productive life cycle. Data were presented as mean values ± SD. **f** Southern blot analysis ($n = 1$) showing a modest increase in HPV16 genome copy number per cell in 3D rafts vs 2D monolayer cells. Strong expression of clinical markers indicative of HPV+, including exportin-5 (**g**) and p16 (**h**) is observed in HPV16+ but not HPV16− epithelium by IHC, counterstained with hematoxylin. Scale bars, 25 μm. Representative images were selected from $n = 1$ raft sets (**g**) or $n = 3$ raft sets (**h**). Source data are provided in the Source Data file.

transcriptomic gene list, we sought to identify "cluster-defining genes" that could serve as biomarkers to independently represent clusters without the need for more complex combinatorial approaches. Candidate cluster-defining genes were identified using three criteria: (1) Expression in at least 80% of cells in the cluster of interest. (2) Robust expression in rafts by RT-qPCR (CT <35) that was limited to the cluster of interest in UMAP feature plot visualization. (3) Availability of commercial RNA-ISH probes and potential for combinatorial detection based on channel availability.

Using this strategy, candidate cluster-defining markers were identified for proof of concept in C2 (C10orf99 and DSG1) and C3 (KLK7, SPRR1A, RDH12, and POF1B) (Fig. 4a). C2 expressed known markers of the stratum spinosum (DSG1) and C3 expressed markers of the stratum granulosum and stratum corneum (POF1B and KLK7, respectively). Importantly, little overlap was observed between cells expressing DSG1 (C2) versus KLK7 (C3) by dual expression plots (Fig. 4b), suggesting that these clusters define distinct subpopulations in rafts. Similar expression levels of C2 and C3 cluster-defining markers were observed in HPV16+ versus HPV16− rafts (Fig. 4c), consistent with the comparable number of cells from these subpopulations in the single cell capture (Fig. 2f). Next, we determined dependence on differentiation of C2 and C3 by comparing the level of gene expression in differentiated specimens (overconfluent cells and rafts) relative to undifferentiated subconfluent cells (Fig. 4d). The data confirmed that, as expected, the terminally differentiated C2 and C3 keratinocyte subpopulations are largely absent in subconfluent monolayer cultures and the epithelial differentiation program is required for their formation. Altogether, we selected KLK7 as a cluster-defining marker for C3 (expressed in 90.6% HPV− C3 cells and 93.2% of HPV+ C3 cells) and termed the C3 population as the terminally differentiated KLK7+ epithelial subpopulation.

RNA-ISH spatial mapping and the C3 biomarker KLK7 were used to locate this differentiated subpopulation in rafts. We discovered an unexpected differential pattern of KLK7 localization in HPV+ versus HPV− rafts, indicative of de-regulation of suprabasal clusters in HPV16+ rafts. While KLK7+ C3 cells were localized to the uppermost layers of HPV16- rafts, HPV16+ rafts harbored an additional compartment above the KLK7+ cells (red bracket, Fig. 4e). These data are consistent with the decreased protein staining previously noted for CARD18 and KLK5 (Fig. 2f) in the superficial epithelial layer and suggest that HPV reprograms the differentiated epithelium to drive the emergence of an HPV-reprogrammed keratinocyte subpopulation. We sought to quantify the localization and thickness of the raft portion above KLK7+ cells in order to demonstrate robust upregulation of this compartment, specifically in HPV+ rafts. To this end, we measured raft thickness from the top of the raft to the top of the KLK7+ cell layer and normalized these values to total raft thickness in each case (Fig. 4f). Three biologically independent rafts were quantified. The KLK7+ cell layer largely extended to the uppermost surface of HPV− rafts (an average of only 5.9% of the raft remaining above), but not in HPV+ rafts where the top third of HPV+ rafts was composed of a distinct superficial compartment (an average of 28.5% of the raft above KLK7+ cells) (Fig. 4g). Thus, these experiments identified an epithelial compartment-specific to HPV16+ epithelium.

**The C9 transcriptome defines a differentiation-dissonant keratinocyte subpopulation that forms the uppermost compartment of HPV16+ rafts**

The C9 subpopulation was greatly enriched in HPV16+ rafts (Fig. 2f) and was uniquely distinct from the remaining 11 clusters by UMAP visualization (Fig. 2e) and in the initial pseudotemporal linear trajectory (Supplementary Fig. 1a), leading to its exclusion. Using our three-step strategy described above, we next identified and visualized candidate cluster-defining C9 biomarkers, which include ELF3, GPR110, and LCN2 (Fig. 5a and Supplementary Fig. 2a). For example, ELF3 was expressed in 100 and 96.6% of HPV16+ and HPV16− C9 cells (pct.1), respectively, and only 21.9 and 6.5% of remaining HPV16+ and HPV16− cells (pct.2), respectively. UMAP visualization confirmed high expressing ELF3 cells to be confined to C9. Relative gene expression by RT-qPCR showed increased expression of C9 markers in HPV16+ versus HPV16− rafts, in line with HPV16-driven enrichment of this population (Fig. 5b and Supplementary Fig. 2b). Additionally, expression of C9 cell markers was increased in models of differentiation (overconfluent cells and rafts), relative to undifferentiated (subconfluent) cells (Supplementary Fig. 2c). These Supplementary Data suggest that although the C9 subpopulation lacks conventional differentiation markers, it is nonetheless dependent upon the differentiation program and must therefore be studied in models of stratified epithelium. RNA-ISH for five C9 cluster-defining markers localized this population to the uppermost suprabasal compartment with robust staining in many cells of HPV16+ rafts; in contrast, HPV16− rafts exhibited weak staining in few cells (Fig. 5c). Quantification of total ELF3 and GPR110 RNA puncta showed increased expression of C9 markers in HPV+ vs HPV− rafts (Fig. 5d). The spatial localization of C9 biomarkers within the stratified squamous epithelium was determined by subdividing rafts into quintiles, from the most basal (Q5) to the most superficial (Q1) layer of the rafts. The distribution of ELF3 (Supplementary Fig. 2d) and GPR110 (Supplementary Fig. 2e) RNA expression was determined across quintiles for three independent rafts. ELF3 (Fig. 5e) and GPR110 (Fig. 5f) expression consistently increased in the suprabasal compartments of HPV+ rafts, most strikingly in the uppermost 20–40% of rafts. Protein levels were examined by IF to determine whether gene expression of C9 markers translates into protein expression in HPV16+ rafts. Consistent with the RNA-ISH results, ELF3 and LCN2 proteins localized to the uppermost compartment of HPV16+ rafts, whereas they were largely absent in HPV16− rafts (Fig. 5g). Altogether, we find that the uppermost suprabasal compartment of HPV16+ rafts lacks gene expression of conventional markers of differentiated stratified epithelia (Fig. 3f), in line with the previously observed suppression of the corresponding proteins (e.g., CARD18 and KLK5; Fig. 3f). Instead, this layer is populated by epithelial cells harboring the unique C9 transcriptome.

The C9 transcriptome was further interrogated to identify candidate targets using pathway analysis and transcription factor enrichment tools. Reactome and KEGG pathway analysis revealed ontology hits, including viral infection, carcinogenesis, keratinization, and immune signaling (Supplementary Fig. 3a, b). Differential expression of HPV16+ versus HPV16− C9 cells showed upregulation of viral and bacterial pathogenesis and intercellular junctions (Supplementary

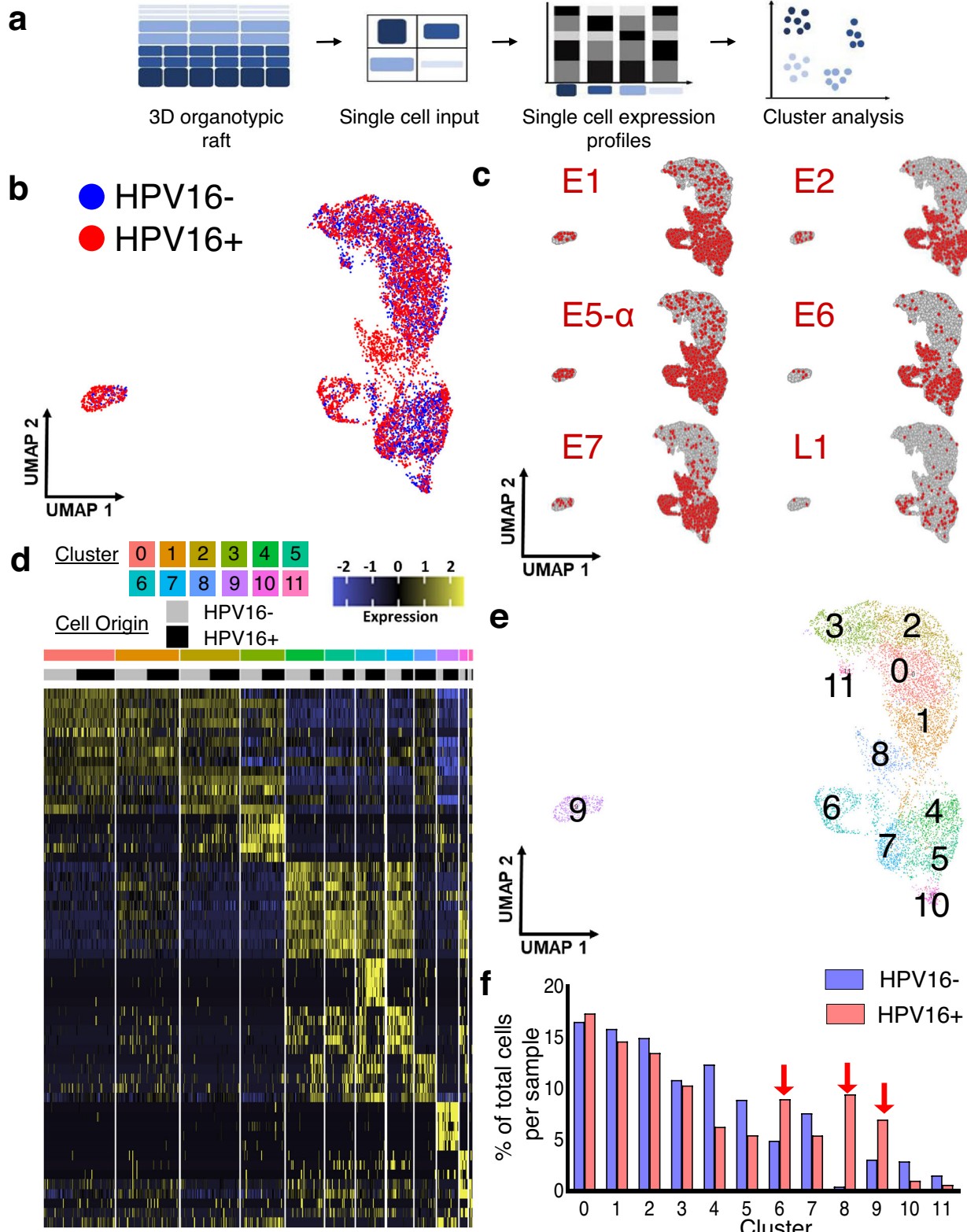

**Fig. 2 | scRNAseq of isogenic HPV16+ and HPV16− epithelium defines 12 distinct keratinocyte subpopulations. a** Schematic of the scRNAseq workflow using 3D epithelial rafts. **b** Combined UMAP of HPV16− (blue) and HPV16+ (red) keratinocytes that were jointly analyzed shows significant overlap following batch correction (*n* = 1 matched set). **c** Feature plots for HPV16 viral gene expression. Unbiased clustering reveals 12 transcriptomically distinct cell populations displayed as a heatmap (**d**) and UMAP (**e**). Gene lists per cluster are provided in Supplementary Data 2. Each keratinocyte subpopulation was assigned a number and color, and the origin of cells (from HPV16− or HPV16+ rafts) was indicated in gray versus black to highlight cluster composition. **f** Distribution of cells in HPV16+ and HPV16− epithelium reveals enrichment of HPV16+ cells in C6, C8, and C9 (arrows). Source data is provided in the Source Data file.

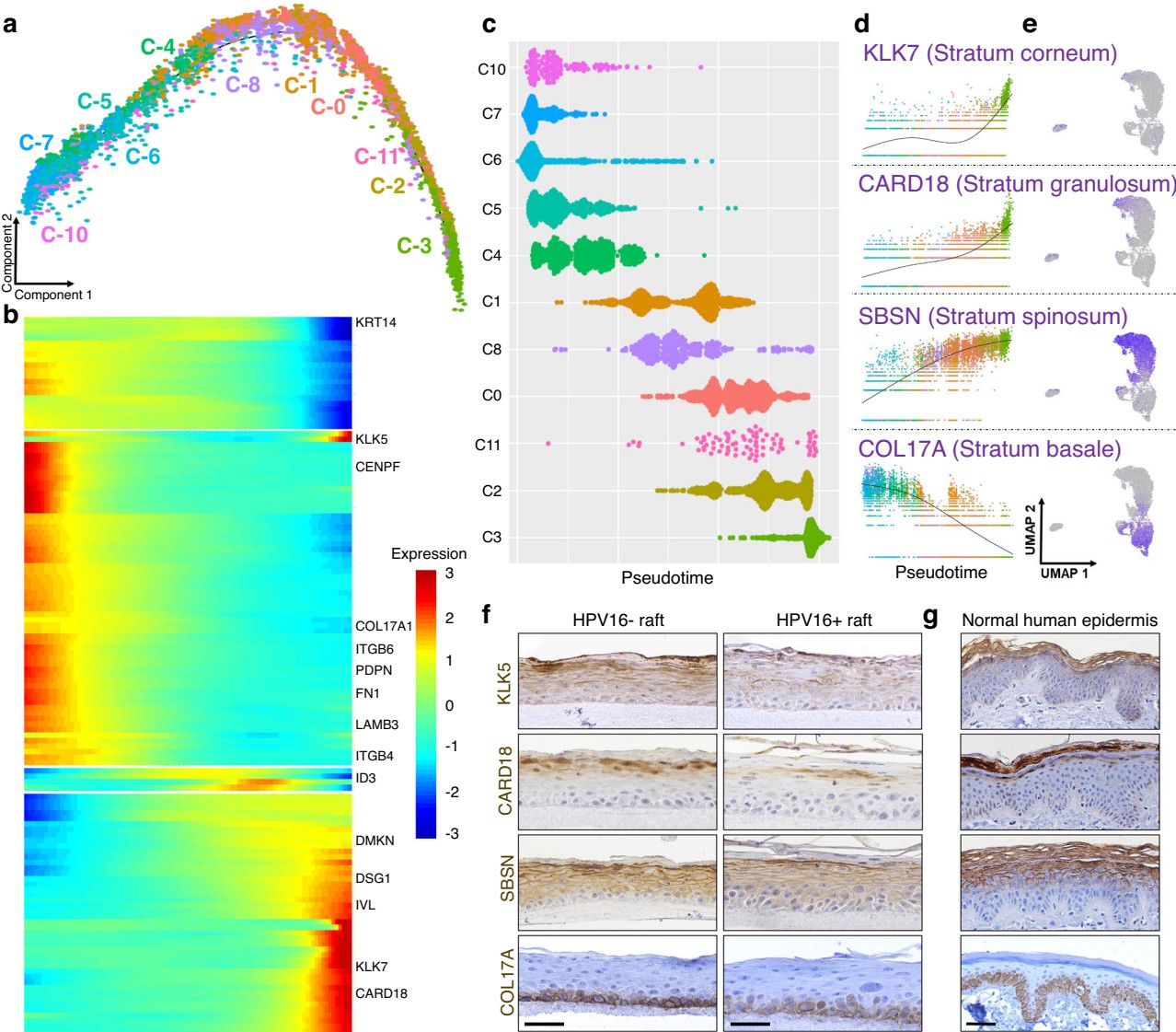

**Fig. 3 | Pseudotime analysis aligns cells along the differentiation program of 3D stratified epithelium. a** Pseudotemporal trajectory analysis, excluding the divergent C9 subpopulation, indicates a linear progression of clusters (*n* = 1 matched set). **b** Pseudotime heatmap with select highlighted genes reflects a progression from basal to differentiated cell types; full list of genes, Supplementary Data 2. **c** The pseudotime violin plot splits the majority of clusters into primarily early (C10, C7, C6, C4, and C5) or late (C1, C8, C0, C11, C2, and C3) stages in the trajectory. Gene expression of established markers of stratified epithelial layers confirms a progression from basal to differentiated populations; shown over pseudotime (**d**) and UMAPs (**e**). **f** IHC staining validates the organization of layers of the stratified epithelium in the HPV16− versus HPV16+ 3D epithelium, with less intense staining in HPV16+ suprabasal layers. Scale bar, 100 μm. Representative image selected from *n* = 10 frames taken per *n* = 3 raft sets. **g** IHC staining of normal human skin is shown for comparison. Scale bars, 50 μm. Representative image selected from *n* = 3 frames taken per *n* = 2 patient biopsies.

Fig. 3c, d) and downregulation of the formation of the cornified envelope (Supplementary Fig. 3e). Together, these ontologies in HPV+ C9 cells indicate the importance for viral pathogenesis and squamous differentiation, albeit not classical differentiation, for the C9 compartment. To capture these features of the C9 transcriptome, we termed the C9 subpopulation HPV-induced differentiation-dissonant epithelial nonconventional (HIDDEN) cells.

### HIDDEN cells are upregulated in HPV+ cervical rafts
To address the possibility that HIDDEN cell detection was an artifact of the NIKS model, we next used organotypic epithelial rafts engineered from primary cervical keratinocytes that were infected with high-risk HPV18 quasivirus versus mock-infected controls (Fig. 6a). HPV+ (versus HPV−) epithelium harbored HPV genomes by DNA-ISH (Fig. 6b) and many intensely stained ELF3+ suprabasal cells by RNA-ISH (Fig. 6c) and

by IF (Fig. 6d). This demonstrates that (1) few ELF3+ cells are observed at baseline in the healthy cervical epithelium, (2) HIDDEN cells are highly upregulated by HPV in the cervical epithelium and (3) other high-risk HPV types such as HPV18 also induce the HIDDEN cell compartment.

### HIDDEN cells are detected at various stages of HPV-driven cervical carcinogenesis
We next sought to evaluate the extent to which our findings in patient-derived cervical rafts translate to primary patient tissues at distinct stages of cervical intraepithelial neoplasia (CIN). CIN lesions, precursors to cervical squamous cell carcinoma (SCC), are ideal for studying HPV infection and progression toward cervical SCCs. Generally, CIN1 lesions harbor viral episomes, whereas CIN2 and CIN3 harbor a mix of episomal and integrated viral genomes with increasing

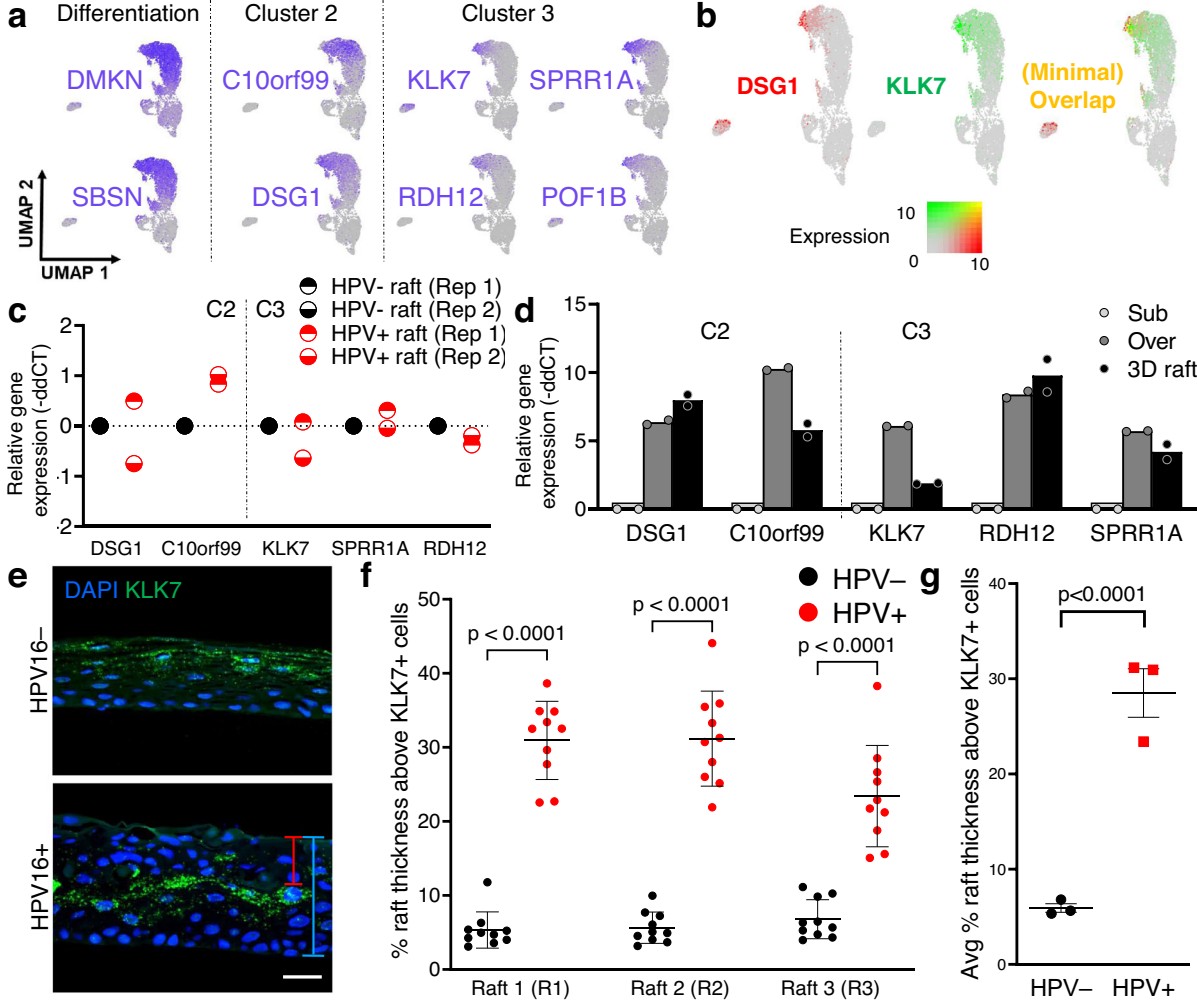

**Fig. 4 | Spatial localization of the terminally differentiated KLK7+ C3 keratinocyte subpopulation in stratified squamous epithelium. a** Feature plots showing differentiation markers and candidate cluster-defining markers for clusters C2 and C3 by UMAP ($n = 1$ matched set). **b** Dual expression plot visualizing cluster-defining markers of C2 (DSG1+) versus C3 (KLK7+) cells, demonstrating minimal overlap of cells with high dual expression. **c** RT-qPCR validation of selected cluster markers confirm similar levels of expression in HPV16− and HPV16+ rafts ($n = 2$). **d** Relative gene expression of C3 and C3 cluster markers in differentiated (overconfluent and raft cultures) vs undifferentiated (subconfluent) HPV− cells validates upregulation of C2 and C3 subpopulations with differentiation ($n = 2$). **e** Localization of the terminally differentiated KLK7+ C3 subpopulation in 3D epithelium by RNA-ISH. The red bracket is representative of the raft thickness measured above KLK7+ cells and the blue bracket of full raft thickness, measures used to quantify the unstained compartment superjacent to C3 KLK7 + cells in HPV16 + epithelium. Scale bar, 25 μm. **f** Quantification of the unstained compartment above KLK7+ cells by relative percent raft thickness (e.g., red bracket relative to blue bracket in **e**) for three sets of HPV ± rafts ($n = 10$ per raft, mean ± SD, two-way ANOVA with Sidak's multiple comparisons, $p < 0.0001$). **g** Average quantification of (**f**) as the mean ± SEM ($n = 3$ rafts, two-way ANOVA, $p < 0.0001$). Source data (**c**, **d**, **f**, **g**) are provided in the Source Data file.

dysplasia and progression to cervical SCC. However, the overall structure of basal and suprabasal epithelial components are retained in CIN1/2 lesions, but not in CIN3 lesions[30].

Cervical biopsies from patients suspected to have CINs were collected, processed, and screened for the high-risk HPV marker p16. H&E stained sections were blindly reviewed for HPV features and CIN lesions by a board-certified pathologist and selected specimens stained by IF for the HIDDEN cell biomarker ELF3. ELF3 protein expression is reported to be low to undetectable in normal ecto-cervical epithelial tissue according to the Protein Atlas[31] and was correspondingly expressed at low levels in cervical tissue appearing morphologically normal and without evidence of HPV infection (Fig. 6e). Consistent with our in vitro models, we found that tissue suprajacent to p16+ basal cells demonstrated strong staining for the HIDDEN cell biomarker ELF3 (Fig. 6f). This included tissues with HPV features and CIN lesions regardless of grade. Typically, ELF3 expression appeared strongest in the uppermost tissue layer, mirroring the pattern observed in rafts. In some areas of tissues with CIN lesions,

ELF3 staining was observed to be more intense in the middle compartment with less intense staining in more superficial cells (representative example Fig. 6f, CIN1). Nonetheless, in all p16+ tissues, HIDDEN cells were greatly amplified. Notably, while established HPV biomarkers (e.g., p16) reflect basal or proliferating cells which must be procured invasively, the ELF3+ HIDDEN cells instead demarcate superficial epithelial cells with the potential of the noninvasive collection.

## HIDDEN cell biomarkers are elevated in HPV+ SCC tumors

Since HIDDEN cells were found to persist through all stages of premalignant cervical tissues (CIN1-3), we proposed that HIDDEN cells further persist from premalignancy into malignant squamous cell carcinomas. In order to evaluate this hypothesis, we analyzed the Pan-Cancer TCGA to determine the expression of HIDDEN cell biomarkers in cancer. We initially focused on HNSCC patient cohorts to directly compare HPV+ versus HPV− tumors; the high prevalence of HPV in cervical SCCs precludes the ability to robustly compare patients by HPV

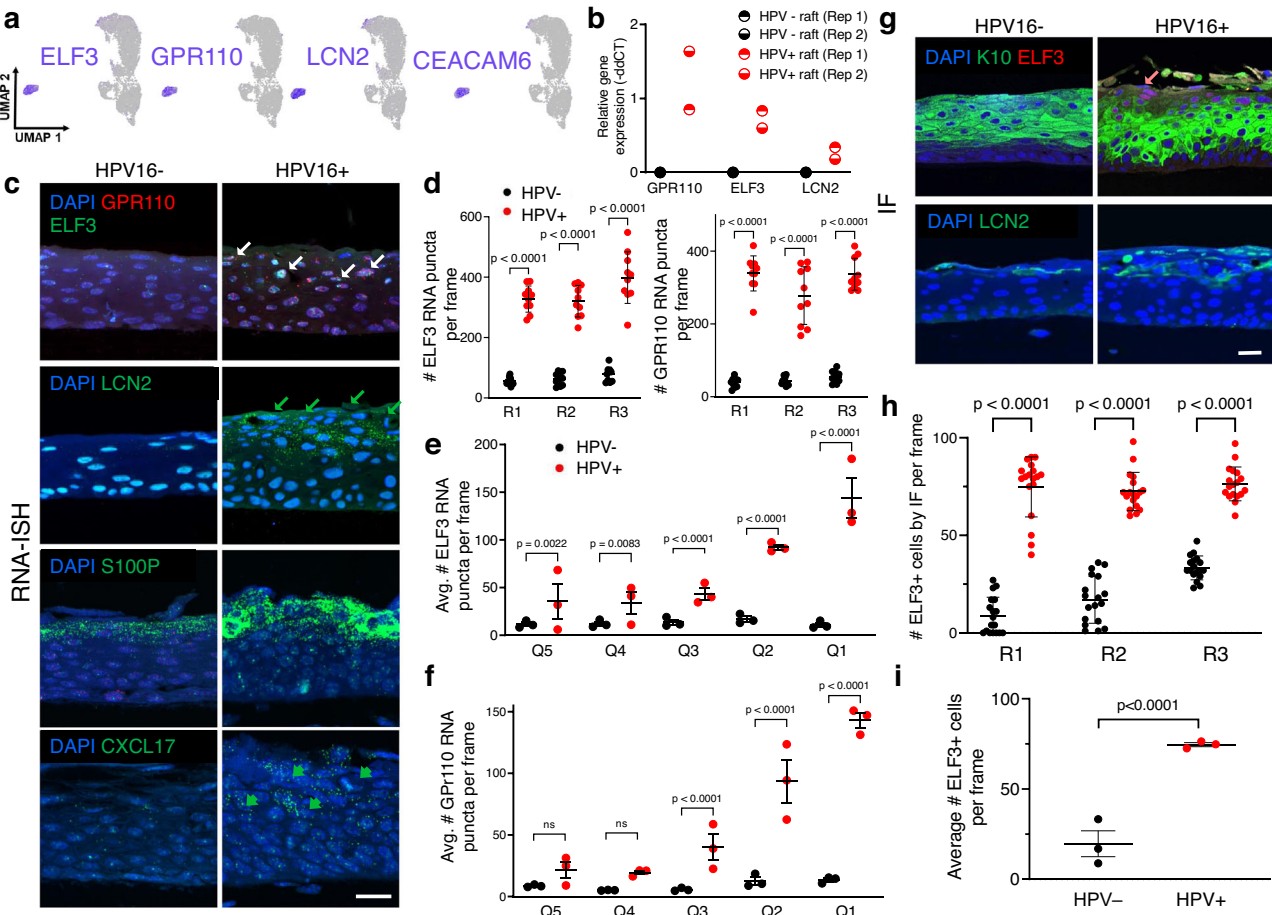

**Fig. 5 | C9 cells occupy the superficial cell layers of the HPV16+ epithelium.**
**a** Feature plots visualize expression of select C9-defining markers ($n = 1$ matched set). **b** Subset of RT-qPCR validation shows increased expression of C9 markers in HPV16+ vs HPV16− epithelium ($n = 2$). Additional C9 biomarker validation is shown in Supplementary Fig. 2B. **c** RNA-ISH validation in 3D epithelium for C9 biomarkers shows many cells with multiple puncta in HPV16+ epithelium, primarily in superficial suprabasal layers, vs very few in HPV16− epithelium. Scale bar, 25 μm. Arrows highlight positive staining in cells. **d** Quantification of total ELF3 or GPR110 puncta in three matched sets of rafts ($n = 10$ frames per raft with data presented as the mean ± SD, two-way ANOVA with Sidak's multiple comparisons, all comparisons $p < 0.0001$) showing consistently higher total number of transcripts in HPV+ vs HPV− rafts. The distribution of ELF3 (**e**) and GPR110 (**f**) expression across rafts by

quintile, where Q5 is basal and Q1 is the uppermost raft fifth, spatially locates C9 marker expression to the most suprabasal raft layers ($n = 3$ rafts presented as mean ± SEM, two-way ANOVA with Turkey's multiple comparisons test, $p$ values indicated per comparison). **g** IF showing a compartment of ELF3+/LCN2+ cells, specifically in HPV16+ rafts that is superjacent to the K10+ differentiated layer and confirms protein-level expression of C9 markers. Scale bar, 25 μm. **h** Quantification of ELF3+ cells by IF across three matched sets of rafts ($n = 10$ frames per raft presented as mean values ± SD, two-way ANOVA with Sidak's multiple comparisons, all comparisons $p < 0.0001$). **i** Average number of ELF3+ cells by IF per frame in HPV+ vs HPV− rafts ($n = 3$ rafts represented as mean ± SEM, two-way ANOVA, $p < 0.0001$). Source data (**b**, **d**–**f**, **h**, **i**) are provided in the Source Data file.

status. Of the 488 HNSCC patients available for analysis, 9% had significantly elevated ELF3 expression ($z \geq 2.0$) compared to the remaining patients. Importantly, this patient sub-group had statistically significant differences in HPV status, International Classification of Diseases (ICD)−10 classification (indicating anatomic location), chromosome 1q abnormalities, and hypoxia scores (Fig. 7a). ELF3 expression was higher in HPV+ vs. HPV− HNSCC (Fig. 7b, $p < 0.0001$), with ELF3 mRNA levels elevated in 24% of HPV+ HNSCCs (z-score ≥2.0) and above average in 75% (Fig. 7c). Conversely, in HPV− HNSCC patients, ELF3 mRNA levels were elevated in only 7% of cases and below average in 67% (Fig. 7c). We next examined ELF3 mRNA levels in HNSCCs grouped by ICD-10 (Fig. 7d). This analysis revealed that, although average ELF3 expression varied across anatomical locations ($p = 0.015$, it was consistently higher in HPV+ vs HPV− samples ($p < 0.0001$). As expected, the distribution of HNSCCs across subsites varied by HPV status: HPV+ HNSCCs predominated in subsites within the oropharynx (e.g., tonsils and base of tongue) while HPV− HNSCCs were more likely to occur in the larynx, lip/oral cavity/pharynx, or tongue. These associations build on our findings in rafts and patient tissue, supporting the presence of an ELF3-driven

differentiated squamous subpopulation in HPV+ disease processes, including cancer.

The remaining significant clinical attributes (1q status and hypoxia) that varied by ELF3 expression were also evaluated. ELF3 mRNA levels correlated negatively with hypoxia scores (Supplementary Fig. 4a). This correlation was unexpected; ELF3 is a known HIF1A target[32] shown to be increased with hypoxia in vitro[33]. While hypoxia is a negative prognostic factor in HNSCC[34,35], its contribution to HPV+ vs HPV− HNSCC is complex. Tumor hypoxia does not appear to vary significantly by HPV status, yet HPV+ patients have improved therapy responses and survival[34,35]. Thus, the observed correlation between ELF3 and hypoxia may be incidental, confounded by other variables, or maybe more complex in vivo than is currently understood. Lastly, since 1q status differed in HNSCCs with high (vs normal) ELF3 mRNA, we assessed whether differences in ELF3 mRNA expression could be accounted for by genome-level abnormalities and/or mutations. As expected[36,37], the overall mutational burden was lower for HPV+ HNSCC than for HPV− HNSCC (Supplementary Fig. 4b). Thus, differences in ELF3 mRNA expression between HPV−

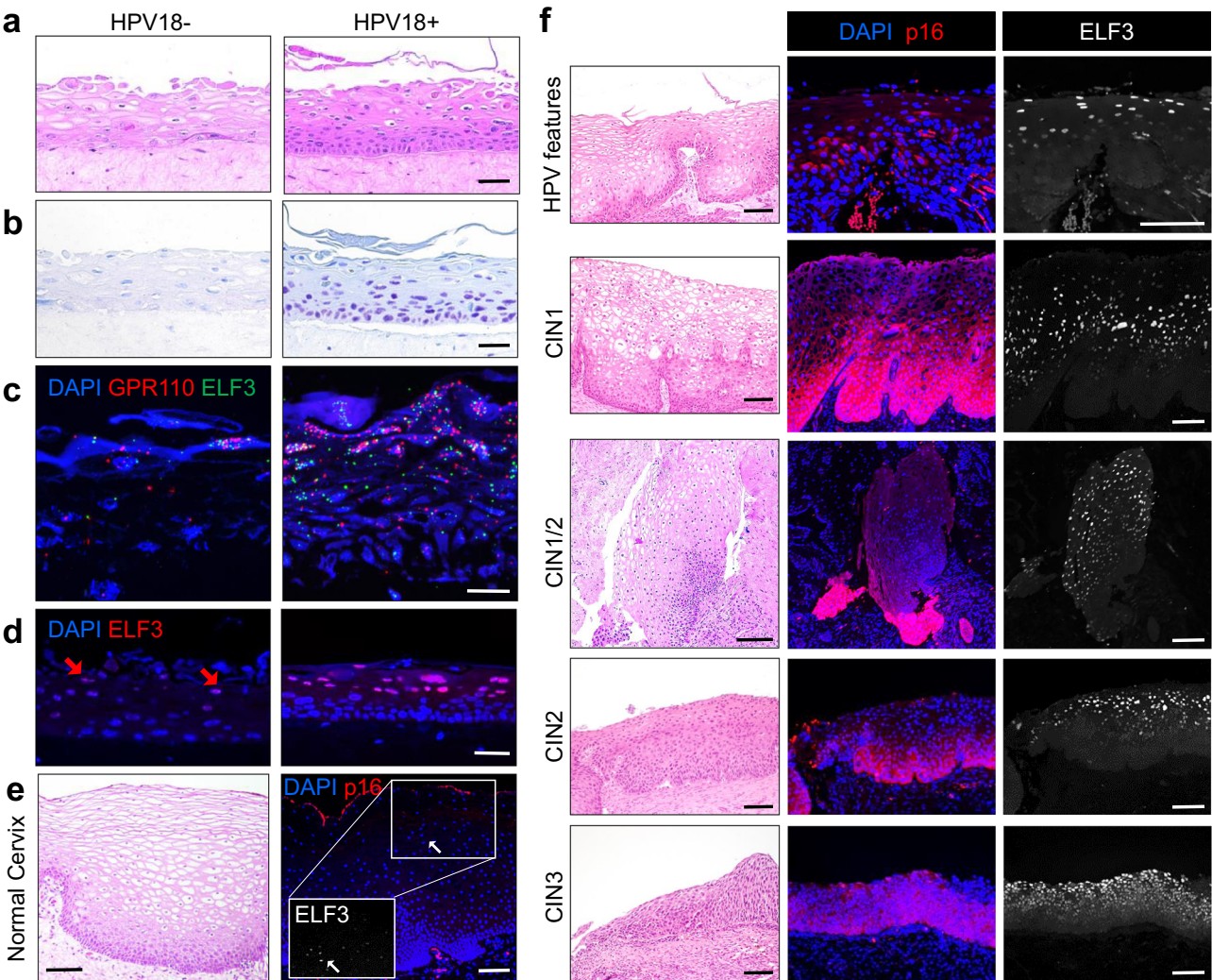

**Fig. 6 | HIDDEN cells are detected in HPV+ cervical rafts and in premalignant cervical lesions. a–d** Representative images from $n = 10$ across each raft set ($n = 1$). **a** H&E of primary human cervical epithelial cells, with or without HPV18, engineered into 3D organotypic rafts. Scale bar, 50 μm. **b** DNA-ISH shows that all cells harbor HPV genomes in HPV+ but not HPV− epithelial rafts. Scale bar, 50 μm. **c** RNA-ISH for GPR110 and ELF3 shows increased puncta in HPV+ vs HPV− epithelium. Scale bar, 25 μm. **d** IF for ELF3 shows many intensely stained nuclei in the suprabasal compartment of HPV+ cervical rafts, with few and weakly stained ELF3+ cells (red arrows) in their HPV− counterparts. Scale bar, 50 μm. **e** Very few, lightly stained ELF3+ cells (arrows) are detected by IF in normal, p16-negative cervical epithelium (white arrow). Scale bar, 100 μm. Representative image from $n = 2$ patient biopsies. **f** Conversely, many strongly stained ELF3 + cells are observed in p16+ tissue harboring features of HPV infection, and in p16+ CIN1-3 lesions, both determined by a board-certified pathologist. Scale bar (H&Es and IFs), 100 μm. Representative images of $n = 2$ patient biopsies from each CIN stage.

and HPV+ HNSCC are unlikely to be explained by mutations in ELF3. ELF3 mRNA levels were also higher in HNSCCs with a gain or amplification of ELF3 copy number, and lower in HNSCCs with a shallow loss (heterozygous) (Supplementary Fig. 4c). Within each of these categories, including normal diploid copy number of ELF3, the expression of ELF3 was higher in HPV+ than in HPV− HNSCC (Supplementary Fig. 4c, $p < 0.001$). Other HIDDEN cell biomarkers, such as MACC1 and GPR110, also had higher levels of expression in HPV+ HNSCCs (Supplementary Fig. 4d). Taken together, we conclude that higher ELF3 expression in HPV+ HNSCC cannot be explained by genome-level differences, but rather reflects HPV-driven mechanisms.

**TCGA data refines the HIDDEN cell signature to core biomarkers shared between HPV-infected and transformed cells**
Large HNSCC and cervical SCC patient cohorts in the TCGA dataset enabled us to refine the HIDDEN cell transcriptome to identify robustly co-occurring biomarkers for the detection of an analogous population in SCCs. We identified 652 genes in HNSCC and 175 genes

in cervical SCC whose expression correlated with ELF3, using a Pearson cutoff of ≥|0.4|(Supplementary Data 5). Overlay of these two gene sets with the HIDDEN cell transcriptome yielded 42 shared genes, including ELF3 (Fig. 7e and Supplementary Data 5). A mutual exclusivity test determined that 17 of the 42 genes co-occurred with ELF3 in patients (Supplementary Table 1). Correlation of these 18 (including ELF3) co-occurring HIDDEN cell biomarkers with patient survival could not be analyzed in the HNSCC patient cohort due to the high variance in ELF3 expression across anatomical sites (Fig. 7d), with each anatomical site having either predominately HPV+ or HPV− tumors.

However, in cervical SCC patients, the high mRNA expression of the 18 co-occurring HIDDEN cell biomarkers was associated with significantly worse patient outcomes by Kaplan−Meier survival curves (Fig. 7f, $p = 0.013$). Taken together, these results from large patient cohorts further substantiate our findings in rafts and patient tissue samples, and provide strong evidence for the existence of a clinically significant epithelial subpopulation that promotes HPV+ carcinogenesis and/or therapy resistance.

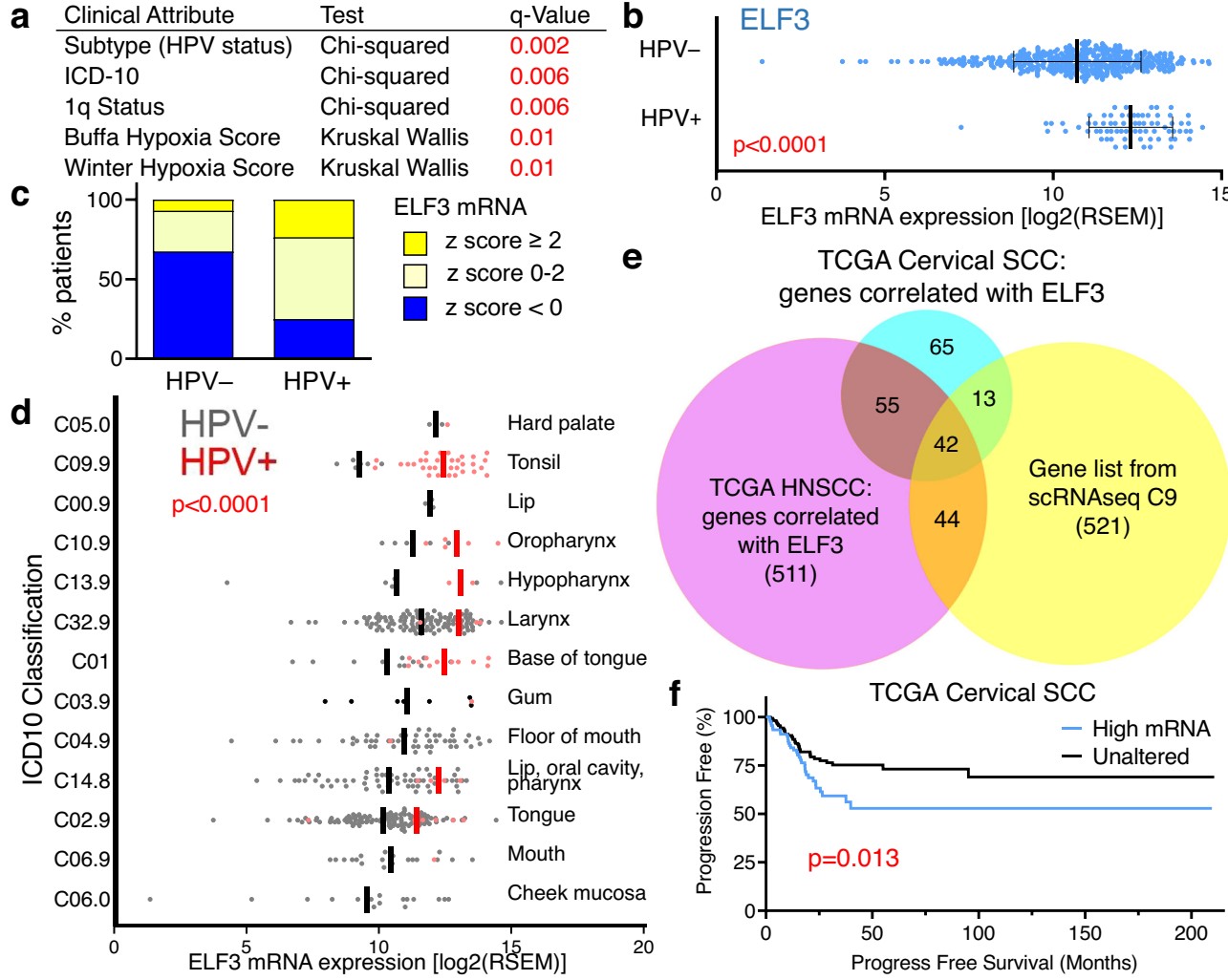

**Fig. 7 | The ELF3-dependent C9 cell population is associated with HPV+ head and neck and cervical SCC. a–d** Analysis of the TCGA PanCancer Atlas for HNSCC (*n* = 488 patients). **a** Table summarizing clinical attributes with statistically significant differences between patients with high ELF3 vs normal ELF3 mRNA expression. **b** ELF3 has increased mRNA expression in HPV+ HNSCC (data were represented as mean values ± SD, Mann–Whitney test, *p* < 0.0001). **c** An increased number of HPV+ HNSCC patients (75%) have above average ELF3 mRNA expression vs HPV- HNSCC (32%). **d** ICD-10 classification of HNSCC labeled by anatomical site with lines representing mean values split by HPV status (two-way ANOVA, *p* = 0.015 between sites and *p* < 0.0001 by HPV status). **e** Venn Diagram showing overlap of the scRNAseq C9 gene list (621 total genes including ELF3) with genes correlated with ELF3 in the TCGA PanCancer Atlas for HNSCC (652 total genes) and cervical SCC studies (175 total genes). Lists are provided in Supplementary Data 5. **f** The Kaplan–Meier survival curve for cervical SCC patients (*n* = 275) with high versus normal mRNA expression of 19 co-occurring markers (Log-rank test, *p* = 0.013). **b–f** Source data are provided in the Source Data file.

## The HPV+ HIDDEN cell signature is detected in published scRNAseq data of HNSCCs

In order to validate the above signature in an independent dataset, we analyzed published scRNAseq data of nine HPV− and six HPV+ HNSCC tumors[38]. This report was exclusively focused on diverse immune cell populations and did not analyze epithelial cells. As described in Materials and Methods, all cells were jointly analyzed, followed by isolation and reclustering of epithelial-derived cells. Cells primarily clustered by tumor specimen as previously reported[38] (Supplementary Fig. 5). To address patient variability underlying this analysis, the batch correction was applied. The resulting clusters of all cells (UMAP by HNSCC HPV status, Fig. 8a, and by clusters, Fig. 8b) demonstrated clustering by cell type, rather than by specimen. Analysis of batch-corrected epithelial cells showed overlap between HPV+ and HPV− HNSCC tumors (Fig. 8c). Using the 18 gene core signature, cells were colored by the number of HIDDEN cell biomarkers expressed (Fig. 8d). Overall, epithelial cells from HPV+ HNSCC demonstrated greater expression than HPV− HNSCC tumors, with the highest number of co-occuring biomarkers concentrated in the C7 subset of cells (circle,

Fig. 8d, e). To confirm that C7 was not derived from a subset of tumors, the proportion of C7 cells from each tumor was determined (Fig. 8f). We found that all but one (HPV− tumor HN5) contributed to C7 and furthermore that C7 was composed of a greater proportion of cells derived from HPV+ vs HPV− specimens. C7 shares approximately one-quarter of genes (89 of 336) with the original C9 gene list (Supplementary Data 2). Altogether, this analysis confirms the increased presence of HIDDEN cells using a published dataset of HPV+ versus HPV− HNSCCs.

## HIDDEN cells are enriched in HPV+ tonsillar rafts and in HPV+ HNSCC patient biopsies

We next tested the presence of HIDDEN cells in HPV-infected and transformed head and neck tissue. The primary tonsillar epithelium was utilized as a model of HPV infection of the oropharynx, the most common site of HPV+ HNSCC development. Tonsil tissue (representative H&E stained section, Fig. 9a) harbored few ELF3+ cells at baseline (Fig. 9b). Tonsillar keratinocytes were derived from two separate donors, nucleofected with HPV16 episomes or maintained as

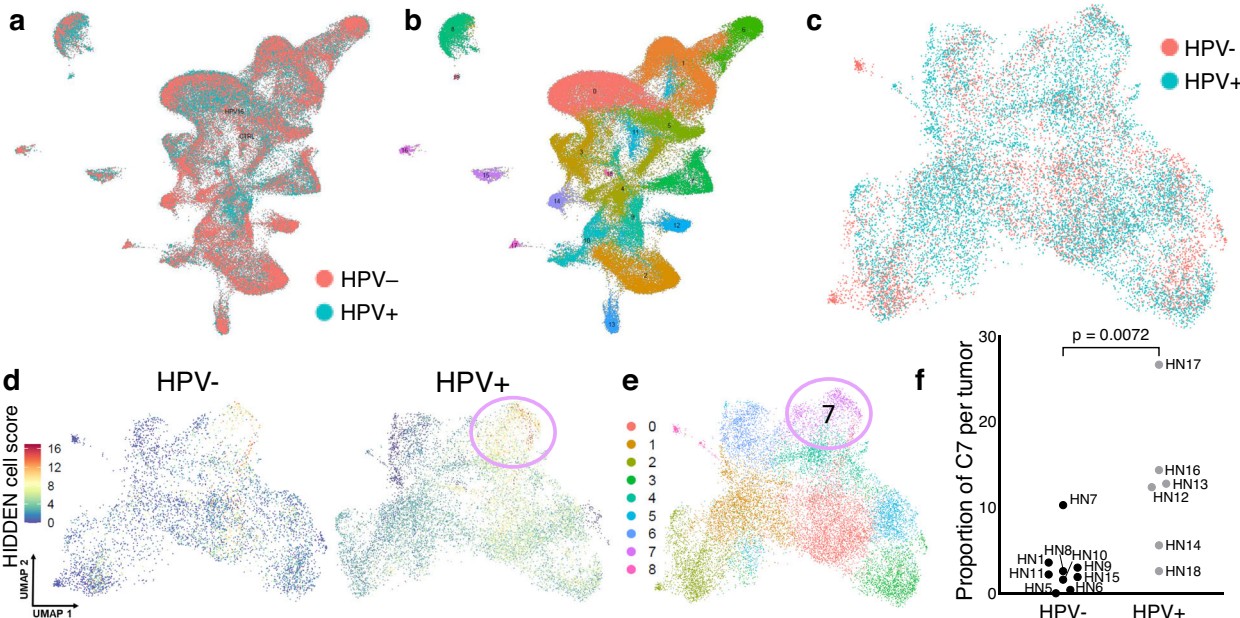

**Fig. 8 | HIDDEN cells are detected in HPV+ vs HPV− HNSCC scRNAseq data.**
**a** UMAP of all cells from published scRNAseq data of nine HPV− and six HPV
+ HNSCC patient biopsies processed through our transcriptomic pipeline.
**b** Transcriptomically distinct clusters, with each color representing a distinct cell
subpopulation. **c** UMAP of keratinocytes only, with cells colored by HPV status of
originating HNSCC. **d** UMAP of HPV− and HPV+ HNSCC epithelial cells with cells
colored by the number of HIDDEN cell biomarkers expressed (18 total) shows

enrichment in HPV+ HNSCCs, particularly in the area indicated with a circle.
**e** scRNAseq clusters of epithelial subpopulations identify C7 as having the highest
expression of HIDDEN cell biomarkers. **f** The proportion of C7 cells from each
tumor demonstrates that all but one (HPV− tumor HN5, 0%) is represented in C7,
with a greater proportion of cells originating from HPV+ tumors (two-tailed
unpaired $t$-test with $n = 9$ HPV− tumors and $n = 6$ HPV+ tumors, $p = 0.0072$). Source
data are provided in the Source Data file.

negative controls, prior to generation of HPV+ and HPV− organotypic
epithelial rafts (Fig. 9c and Supplementary Fig. 6a). DNA-ISH for high-
risk HPV confirmed the expected HPV status (Fig. 9d). RNA-ISH for
ELF3 and GPR110 (Fig. 9e and Supplementary Fig. 6b) and IF for ELF3
(Fig. 9f and Supplementary Fig. 6c) validated significant upregulation
of HIDDEN cells with HPV infection in tonsillar rafts derived from both
donors.

Next, HNSCC patient biopsies were probed for gene and protein-
level expression of HIDDEN cell biomarkers. Many cells with multiple
puncta were detected in two cases of p16+ HNSCCs by RNA-ISH but
were largely absent in two cases of p16-negative HNSCCs (Fig. 9g).
Similarly, ELF3 protein was robustly detected in HPV+, but minimally in
HPV−, HNSCCs with HPV status confirmed by p16 staining (Fig. 9h).
These Supplementary Data Suggest HIDDEN cells persist through
stages of HPV-driven carcinogenesis into SCCs. Altogether, we show
enrichment of HIDDEN cell biomarkers in patient-derived 3D in vitro
models of cutaneous, cervical, and tonsillar epithelium, as well as in
HPV-driven premalignant and malignant patient biopsies that reflect
stages of cancer evolution.

## ELF3 is a potential master regulator of HIDDEN cells

The C9 gene list (Supplementary Data 2) was next utilized to identify
candidate master regulators of HIDDEN cells. The HIDDEN cell tran-
scriptome was initially analyzed by Transfac for putative shared tran-
scription factor binding sites (Supplementary Fig. 7). Each candidate
transcription factor motif was plotted based on statistical significance,
with multiple motifs identified for some transcription factors (Sup-
plementary Fig. 7, genes above dotted line). These potential master
regulators included HPV gene targets (e.g., the *E2F* family as a target of
E7), challenging candidates for inhibition studies due to ubiquitous
expression in many tissues (e.g., *MAZ*[39] and SP1[40]), and genes important
for basal keratinocyte functions (e.g., *KLF* family)[41] that were not
expressed exclusively in HIDDEN cells. Such characteristics suggested
a lack of specificity and, ultimately, safety for the targeting of these

candidates to suppress the HIDDEN cells and eliminated consideration
for further mechanistic studies. *ELF5*, another member of the ETS
transcription factor family with an identical binding site to *ELF3*[42,43],
was identified as a potential master regulator in this analysis. ELF3 is a
known regulator of at least one other verified HIDDEN biomarker,
LCN2[44], and, importantly, its expression was limited to C9 by UMAP
visualization (Fig. 5a). We, therefore, selected *ELF3* for further analysis.
The list of transcription factors was additionally evaluated for binding
sites, specifically within *ELF3*, representing potential upstream reg-
ulators. These hits, indicated by red arrows (Supplementary Fig. 7),
included *NF-kB*, a known regulator of *ELF3*[45].

Advanced analysis of promoter regions of the C9 gene list were
next performed using two complementary methods: HOMER[46] to
assess motif enrichment and RELI ChIP-seq peak enrichment[47] to
estimate the significance of transcription factor binding intersections
in these regions against a library of 11,054 functional genomics data-
sets. These analyses yielded a ranked list of potential transcription
factors that may drive HIDDEN cell formation. The top 25 results were
manually reviewed (Supplementary Data 6). Importantly, *ELF3* was in
the top 97th percentile of hits (#22 of 752, Fig. 10a summary, Fig. 10b
ELF3 motif). Thus, *ELF3* was independently identified as a top candi-
date transcriptional regulator of the HIDDEN cells.

## ELF3 knockdown suppresses the HIDDEN cells in HPV16+ epithelial rafts

To test whether ELF3 is required to sustain HIDDEN cells, ELF3 was
knocked down in HPV16+ NIKS and used to generate rafts. H&E stain-
ing of vector control and ELF3-depleted rafts showed a more uniform
presence of the superficial granulosa and keratinized corneal cell lay-
ers that was variable across rafts, being most prominent in the ELF3sh-
2 knockdown rafts (Fig. 10c). ELF3 knockdown was confirmed by IF
staining for ELF3 (Fig. 10d) and by western blot analysis (Fig. 10e). Total
ELF3 protein levels were elevated in HPV+ versus HPV− rafts and
greatly reduced in rafts generated from knockdown cells (Fig. 10d, e).

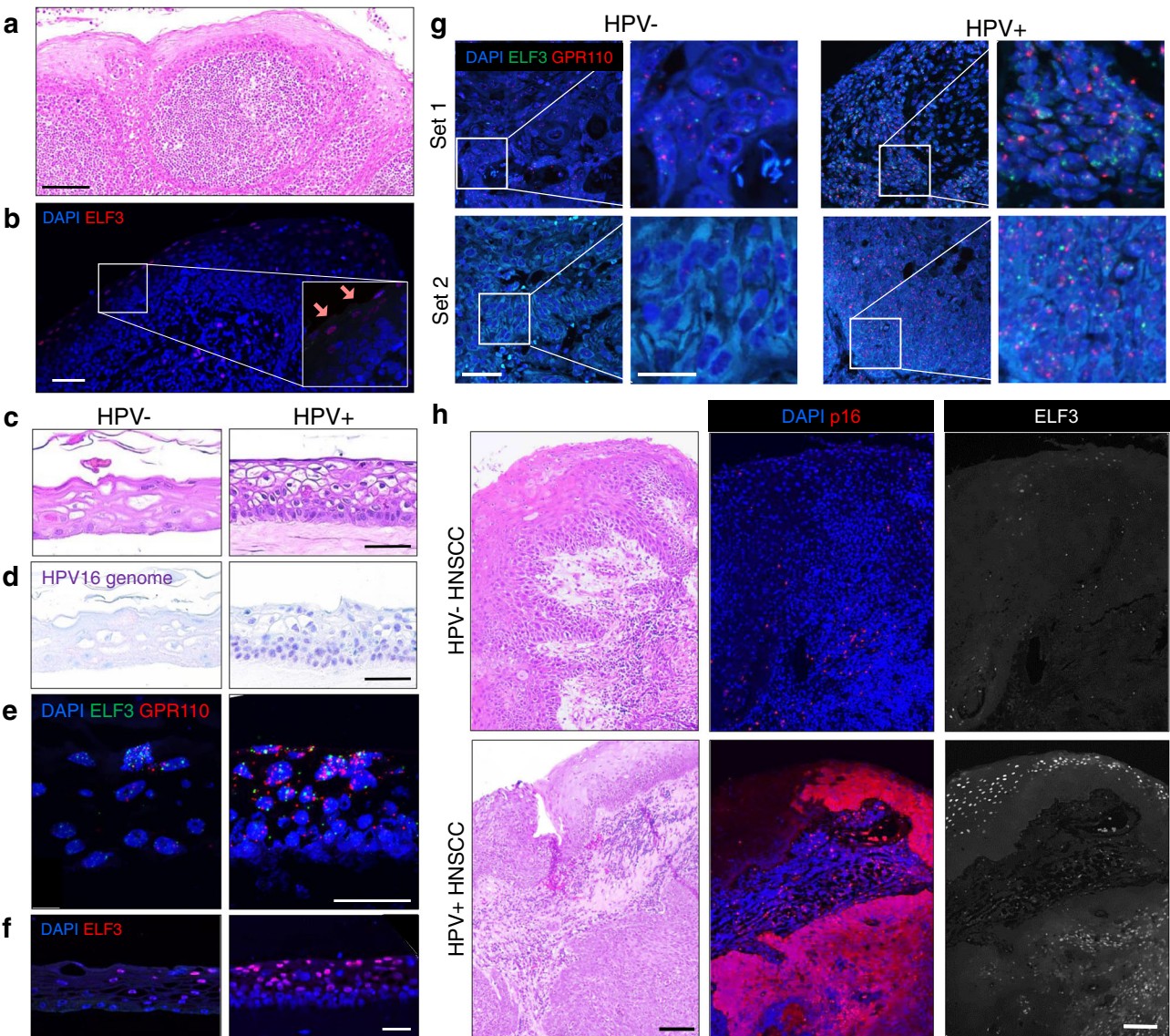

**Fig. 9 | HIDDEN cells are detected in HPV+ tonsil-derived epithelium and in HNSCC patient biopsies. a** Representative H&E from *n* = 5 frames of the patient tonsil (Donor A) that was used to culture patient-derived tonsil keratinocytes to support experiments. Scale bar, 100 μm. **b** Representative IF showing few cells with light ELF3+ staining (arrows in inset) in tonsil tissue, indicating the relative absence of HIDDEN cells. Scale bar, 100 μm, *n* = 5 frames across tonsil tissue. **c**–**f** Representative images from *n* = 10 frames from a raft set generated from Donor A. **c**–**f** Scale bar, 50 μm. **c** Representative H&Es of epithelial rafts generated from tonsil-derived keratinocytes from Donor A that were either nucleofected with HPV16 or not nucleofected. Many more basal cells are seen in HPV+ tonsil

epithelium. Scale bar, 50 μm **d** DNA-ISH for HPV genomes confirms HPV status of 3D rafts. **e** RNA-ISH for ELF3 and GPR110 shows few cells with puncta in HPV− rafts, and many puncta in HPV+ rafts. **f** IF staining reveals significant upregulation of ELF3+ HIDDEN cells in the suprabasal compartment. **g**, **h** Representative images from *n* = 10 frames from two p16+ and two p16− HNSCC biopsies. **g** RNA-ISH for HIDDEN cell biomarkers ELF3 and GPR110 shows many puncta in HPV+ vs HPV− HNSCC patient biopsies. Scale bar, 50 μm in the main image and 20 μm in an insert. **h** Representative images of HNSCC tumors stained by H&E stain and by IF. Stain for p16 confirms the HPV status of tumors and ELF3 staining demonstrates protein-level expression of HIDDEN cells, specifically in HPV+ HNSCCs. Scale bars, 100 μm.

Notably, ELF3 knockdown almost completely eliminated ELF3+ HIDDEN cells in the uppermost compartment of HPV+ rafts, suprabasal to the K10+ layer (Fig. 10f). Consistent with the loss of the ELF3 cellular compartment, there was reduced expression of genes co-expressed with ELF3 in the HIDDEN epithelial subpopulation including GPR110, CEACAM6, MACC1, LCN2, and PSCA (Fig. 10g). These data demonstrate that the ELF3 transcription factor is required for the formation and/or maintenance of the suprabasal HIDDEN cell compartment in stratified HPV16-infected epithelium.

## Discussion
In this study, we used scRNAseq to define keratinocyte subpopulations in HPV16+ versus HPV16− epithelium, and mapped select

epithelial subpopulations within the geography of 3D stratified squamous epithelium. We chose the well-established HPV16-isogenic NIKS organotypic epithelial raft model[48–50] that mimics squamous epithelial morphology with basal and successively more differentiated suprabasal epithelial layers present in human squamous tissues[51]. NIKS rafts have been extensively used to uncover mechanisms of squamous differentiation[52,53], HPV maintenance and replication[48,54,55], viral gene expression[49,56,57], and HPV-induced carcinogenesis[58–60]. This manipulatable isogenic model system captures the effects of HPV infection on epithelial biology in the absence of confounding variables such as host-dependent immune and environmental exposures to discover cell-autonomous mechanisms critical for viral persistence.

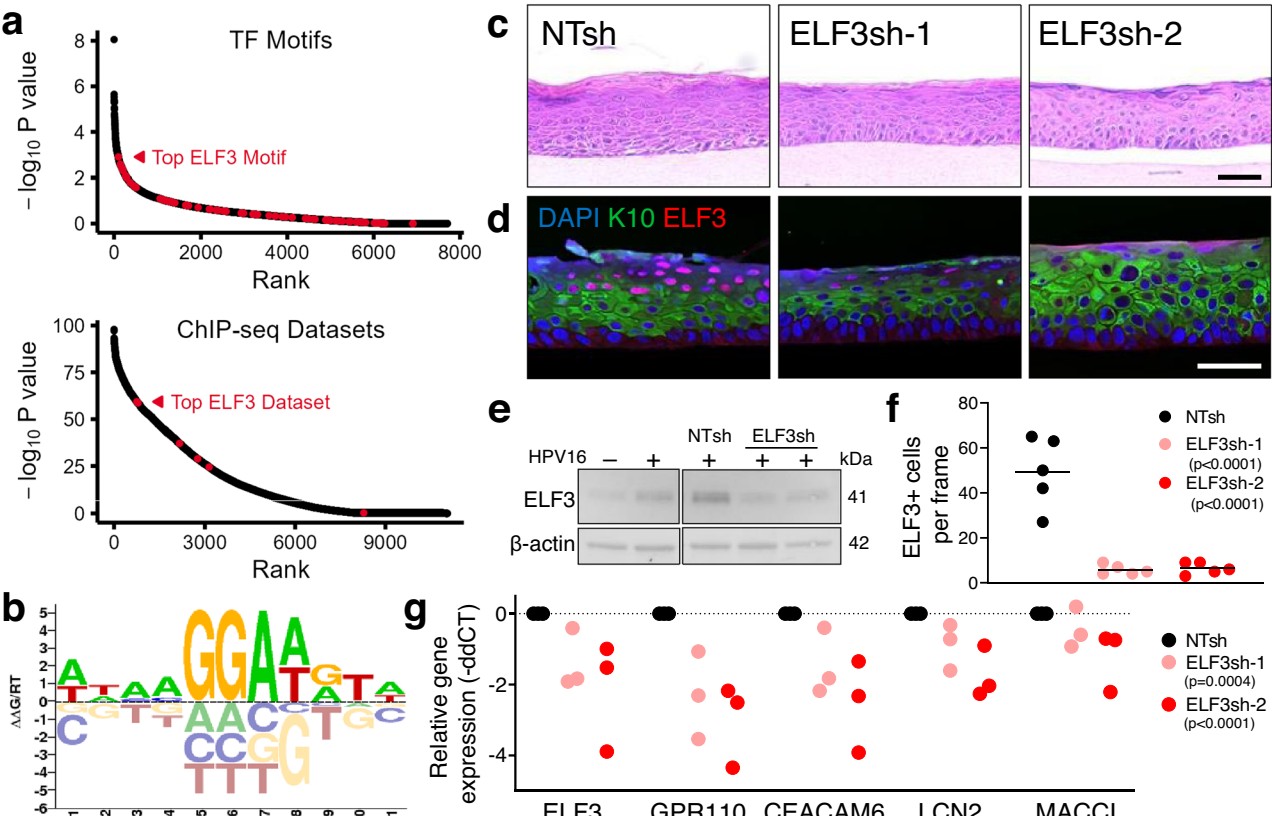

**Fig. 10 | The HIDDEN cell compartment requires ELF3 expression. a** Summary of RELI (top) and HOMER (bottom) analysis indicating ELF3 hits in red (permutation-based strategies detailed in Methods.) Source data is provided in Supplementary Data 6. **b** The ELF3 motif was identified by HOMER and RELI analysis as a potential driver of HIDDEN cell formation. **c** Representative H&E images from *n* = 1 raft set generated from HPV16+ cells transduced with NTsh or ELF3sh constructs. Scale bar, 50 μm. **d** Representative IF image from *n* = 1 raft set of HPV16+ NTsh rafts and ELF3sh rafts validates a dramatic decrease in ELF3+ HIDDEN cells, localized on the surface epithelium, with ELF3 knockdown and elimination of the ELF3+ HIDDEN cell compartment suprabasal to the K10+ layer (representative of *n* = 5 frames across 1 set of rafts). Scale bar, 50 μm. **e** Western blot of 3D raft lysates shows increased ELF3 protein levels in HPV+ vs HPV16− rafts, and a decrease in ELF3 in ELF3sh knockdown vs NTsh control rafts. **f** Quantification of the number of ELF3+ cells per frame, (*n* = 5 per raft with lines representing mean values, two-way ANOVA with Dunnett's multiple comparisons against NTsh, *p* < 0.0001 for ELF3sh-1 and ELF3sh-2). **g** RT-qPCR of HIDDEN cell biomarkers is consistently decreased in ELF3sh rafts compared to NTsh rafts (*n* = 3 rafts, two-way ANOVA with Dunnett's multiple comparisons against NTsh, *p* < 0.0001 for ELF3sh-1 and ELF3sh-2). **f, g** Source data is provided in the Source data file.

Although widely used for studies of HPV, important limitations of this system include the immortalized nature of the NIKS, a duplication of the long arm of chromosome 8, the relative absence of productive viral amplification, which was reflected in our scRNAseq data, and a male origin from human foreskin. While high-risk HPV-associated penile cancer can originate in the foreskin, the incidence of penile cancer is rare when compared to cervical or head and neck cancer. We, therefore, also included HPV+/- isogenic primary tonsillar and cervical organotypic epithelial rafts to validate key findings, thereby extending our data to primary mucosal rafts that are relevant to cervical and head and neck SCC.

The scRNAseq analysis described here identifies 12 transcriptomically distinct keratinocyte clusters, which now enables studies to precisely define the molecular reprogramming of HPV+ compared to HPV− naïve epithelium. Batch correction aligned analogous keratinocyte subpopulations to highlight differences in relative abundance and presence of specific subpopulations and to enable direct differential expression analysis between HPV-infected and uninfected cells per epithelial cell subtype. HPV reprogramming of each epithelial subpopulation forming the conventional basal and suprabasal epithelial layers could thus be assessed.

Using the scRNAseq atlas, we identified differences in the composition of the suprabasal compartment of HPV+ epithelium. Namely, we found that HIDDEN cells typically form the surface

layers of HPV16-infected rafts and are marked by the expression of a broad array of genes implicated in stemness (e.g., *KRT15*[61,62] and *KLF5*[41]), terminal differentiation (e.g., *TJP3*[63]), and carcinogenesis (e.g., *CEACAM6*)[64–66]. These findings extended to primary human models of the cervical epithelium (+/-HPV18) and tonsillar epithelium (±HPV16), indicating the importance of HIDDEN cells in high-risk HPV-driven diseases at multiple epithelial subsites. Although HIDDEN cells were differentiation dissonant in the pseudotime trajectory (Supplementary Fig. 1a), their localization within the superficial layers of stratified squamous epithelium (Fig. 5e, f) and formation in differentiated but not undifferentiated models (Supplementary Fig. 2c) support dependence of this compartment on a differentiated environment as a fundamental part of the HPV life cycle. We further identified the epithelium-specific ESE (ETS) transcription factor *ELF3* as an HPV effector that is required to form or sustain the HIDDEN compartment. Elevated ELF3 activity induced by HPV16 may be reflective of possible links to cancer development or progression.

Despite a substantial body of research on HPV infection, there is a paucity of studies that specifically analyze epithelial cells in HPV+ vs HPV− HNSCCs which would have identified the C9/HIDDEN cell signature. The existing literature is limited by the frequent exclusion of epithelial cells prior to scRNAseq, by unknown HPV status, and by the overwhelming focus on immune-related processes. Identification of

the epithelial HIDDEN cell transcriptome will now enable mechanistic and functional studies in the field.

Detection of HIDDEN cells upregulated in p16+ CINs and HPV+ HNSCCs implies that these cells are maintained and continuously upregulated by HPV throughout the progression from HPV-infected epithelium to dysplastic preneoplasia and cancer. TCGA analyses demonstrated that high ELF3 expression levels were associated with HPV+ head and neck cancers regardless of the subsite, and co-occurring HIDDEN cell biomarkers were preliminarily associated with worse outcomes in cervical cancer. Outcomes by ELF3 expression could not be similarly determined in HNSCC TCGA data due to the paucity of HPV+ and HPV− specimens occurring at the same anatomical site. Thus, the association between HIDDEN cell upregulation and clinical outcomes requires additional study. Nonetheless, while a functional role for HIDDEN cells in human SCC development is not yet clear, enrichment of this subpopulation in high-risk HPV-infected organotypic rafts, primary tissues, and published scRNAseq experiments supports pro-carcinogenic activities.

Multiple scenarios could be at play for a hypothesized role for HIDDEN cells in worse outcomes for HPV+ cancers. HIDDEN cell effects on basal cells might occur via paracrine activities that create a pro-tumor microenvironment, and particularly in severely dysplastic lesions where ELF3+ HIDDEN cells appear to expand downwards into the basal layer (Fig. 6f). Such activities may act on aspiring malignant cells to promote proliferation or survival, or may reprogram the tumor microenvironment to promote angiogenesis or suppress immune surveillance. Alternatively, the HIDDEN keratinocyte signature harbored nonconventional differentiation markers, including select mucins and enzymes involved in o-linked glycosylation (Supplementary Fig. 3a). HIDDEN cells might therefore provide a physical barrier against exogenous anti-viral or anti-tumor responses. Lastly, HIDDEN cells featured gene markers characteristic of primitive stem/progenitor cells, such as KLF5 and K15. It is, therefore, also possible that these cells harbor quiescent stem-like features which might convert to a proliferative state during tumor evolution. Such potential roles remain to be identified.

Altogether, our studies (1) report the transcriptomic landscape of high-risk HPV-infected stratified epithelium, (2) discover epithelial HIDDEN cells that form a compartment and have the potential to mark the surface of HPV-infected tissue, (3) identify ELF3 as a required HPV-induced transcriptional driver of the HIDDEN compartment, and (4) detect and associate HIDDEN cell enrichment in HPV+ SCCs with poor prognosis.

## Methods
All studies comply with all relevant ethical regulations. All research protocols involving human subjects were subjected to review by Institutional Review Boards at the University of Cincinnati, Cincinnati Children's Hospital Medical Center, or the University of Arizona, as detailed in the relevant study methods below, and received approval for the study protocols.

### Monolayer cell culture
HPV+ and HPV− near diploid immortalized keratinocytes that form a skin (NIKS) were provided by the Lambert laboratory, confirmed with STR profiling, and cultured on irradiated murine fibroblasts (J2-3T3)[51]. These cells were either used to generate organotypic rafts or were harvested at subconfluency (40–50% confluency) or overconfluency (48 h past confluency)[67]. Mycoplasma testing was performed regularly and cell lines were confirmed to be negative. Cells were provided by the Lambert lab and confirmed by STR profiling.

### Generation of 3D organotypic rafts from NIKS
3D organotypic epithelial rafts were generated using validated protocols[68]. Briefly, $1 \times 10^6$ NIKS were plated on porcine collagen matrix harboring embedded murine J2-3T3 fibroblasts. After 4 days, rafts were transitioned to an air-liquid interface and high calcium medium to generate stratified epithelium. The medium was then changed every other day for 14 days, at which point rafts were harvested, as appropriate, for subsequent analysis. For embedding, 3D rafts were fixed in 4% PFA for 30–60 min at room temperature, dehydrated, embedded in paraffin, and sectioned at 5 μm thickness onto acid-treated SuperFrost Plus glass slides (Cardinal Health, Dublin, OH, USA M6146-PLUS). After physical removal of the underlying collagen layer, rafts were rapidly dissociated using 0.25% trypsin and mechanical disruption, quenched with medium, filtered through 35 μM cell strainers (Corning, Corning, NY, USA 352235) to remove debris, and confirmed to be a single cell suspension with viability above 70% as assessed by Trypan Blue staining.

### Immunohistochemistry (IHC) and IF
Antibody information, including dilutions and validations, are detailed in Supplementary Data 1. Tissue sections were deparaffinized and antigen retrieval was performed in 1X RNAscope Target Retrieval Reagent (Advanced Cell Diagnostics, Inc., Newark, CA, USA 322000) in a Decloaking chamber (Biocare Medical, Pacheco, CA, USA DV2004MX) for 15 min at 110 °C and 6 psi. IF staining was performed[67]. For IHC, the Vectorstain Immunodetection kit (Vector Laboratories, Burlingame, CA, and HOH-3000) was used with modifications[69–71]. In brief, following antigen retrieval, slides were treated in 0.3% hydrogen peroxide for 30 min to eliminate endogenous peroxidase activity. Non-specific antibody binding was inhibited by incubating sections with a serum-free protein-blocking solution. Tissue sections were incubated with primary antibody for 1 h at room temperature or overnight at 4 °C (p16 only). A secondary biotinylated multilink antibody was applied for 1 h, followed by streptavidin-peroxidase incubation for 30 min. Using freshly prepared 3.3'-diaminobenzidine (Vector Laboratories, Burlingame, CA, SK-4100) and DAKO Liquid DAB Substrate-Chromogen solution as a chromogen, the enzymatic reaction was visualized. Slides were counterstained with hematoxylin.

### RNA-ISH
mRNA expression was detected using the RNAscope® 2.0 Assay (Advanced Cell Diagnostics, Inc., Hayward, CA, USA)[72]. Selected probes for human tissue are detailed in Supplementary Data 1.

### RNA-ISH quantifications
For quantification, Z-stacks were converted from. nd2 to. ims format using the Imaris converter software and opened in Imaris. To define the organotypic raft height, the Imaris "surface creation tool" was used to manually demarcate the top of the organotypic raft and spot creation tool was used to indicate the bottom of the organotypic raft, the average height of the raft was measured using the shortest distance from the surface (top of the raft) to the bottom of the raft. The spots tool was used to outline the region above (cells <5 KLK7 puncta) the KLK7 expressing cells. The average distance was measured using spot statistics for the "shortest distance to the surface" (top of the raft) to the top of the KLK7 expressing cells. This distance was plotted as a percentage of the total raft height.

To quantify individual ELF3 or GPR110 RNA-ISH signals, the height of the raft was determined as described above. The raft height was divided into five equal quintiles, where Q5 was the bottom of the organotypic raft and Q1 was the top. The Imaris "spots" detection tool was then used with an estimated "XY diameter" for spot detection of 0.5 μm to enable the measurement of puncta. "Spots" were created for both ELF3 and GPR110 channels. The number of spots in each quintile was quantified per frame, with ten frames representing each raft. The total number of spots was summed per frame and reported, as well as the breakdown of spots per quintile to indicate spatial localization.

## Quantitative RT-PCR

RNA was isolated using the RNAeasy Mini Kit (Qiagen, Valencia, CA, USA) in parallel sets of HPV16- and HPV16+ NIKS cell pellets from 2D cultures, cells from dissociated rafts, and intact rafts after removal of the underlying collagen. For intact rafts, tissue was consecutively disrupted manually using an 18 and then 21-gauge needle (Thermo Fisher Scientific, Waltham, MA, USA, 305167, 305196). The quality of extracted RNA was confirmed by NanoDrop to have 260/280 and 260/230 ratios between 1.8–2.0. A 1 μg aliquot of each sample was reverse transcribed into cDNA using the QuantiTect Reverse Transcription kit (Qiagen, Valencia, CA, USA, 74106) according to the manufacturer's instructions. Primers specific to each gene (Supplementary Data 1) were designed using Primer-BLAST[73] and purchased from IDT(IDT, Coralville, IA). RT-qPCR was performed on an Applied Biosystems Step One Plus real-time PCR Stem with StepOne software v2.2 (Applied Biosystems, Carlsbad, CA, USA). Assays were performed in accordance with the manufacturer's instructions. RT-qPCR data were analyzed by calculating the ΔΔCt of genes of interest relative to the housekeeping gene β-actin.

## Southern blot analysis

Total genomic DNA was extracted from parallel sets of HPV16- and HPV16+ NIKS cell pellets from 2D cultures and 3D rafts using Qiagen's DNeasy Blood and Tissue kit. A 5 μg DNA aliquot was digested with Bam HI to linearize the HPV16 genome for the purpose of determining viral genome load. Digested genome sequences were electrophoresed on a 0.8% agarose gel along with an HPV16-containing plasmid that had been digested with Bam HI to release the viral genome as a standard. After electrophoresis, DNAs were transferred to the Hybond N+ membrane (Amersham). The membrane was then probed with a set of 20 oligos complementary and specific to HPV16 that had been labeled with γ−32P-ATP. To visualize HPV16 DNA, the washed membrane was exposed to a PhosphorImager screen that was then scanned using a Typhoon laser-scanning platform (GE Healthcare).

## DNA-ISH for HPV

HPV DNA was detected by ISH by the CCHMC Pathology Research Core[74]. In brief, a chromogenic ISH test for high-risk HPV genomes, including HPV16, was performed (Ventana Medical Systems). Sections were incubated with a fluorescein-tagged DNA probe (Enzo Life Sciences, Farmingdale, NY, ENZGEN1146000) and counterstained using the fully automated Ventana BenchMark instrument.

## Single-cell RNA sequencing

For 3′ scRNAseq, matched HPV16- and HPV16+ rafts generated from isogenic NIKS lines were harvested on day 14 post-plating, dissociated into single cells separately but in parallel, and subjected to 10x Genomics Chromium genomic sequencing. A total of 3833 HPV16- and 4275 HPV16+ dissociated cells were included in the analysis. There was an average of 20,619 reads per cell from HPV− rafts and 29,537 reads per cell from HPV+ rafts. Five sets of biological replicates were dissociated in parallel, snap-frozen, and stored at −80 °C for subsequent validation. HPV16− and HPV16+ raft transcriptome datasets were then combined for unsupervised clustering using the Seurat R package[75,76]. Fastq files were processed through the 10x pipeline, CellRanger v3.0.2, to obtain a gene expression matrix using a custom-generated human genome, hg19, that contained annotated sequences for HPV16. To maximize cell quality, cells expressing <1000 unique human genes and/or >20% mitochondrial genes were excluded from subsequent analysis. The gene expression matrix was normalized to 10,000 molecules per cell and log-transformed. The batch correction was performed by integrating the Supplementary DataSet based on anchors, and shared cell types between the two Supplementary DataSets. This was done by using the FindIntegrationAnchors and

IntegrateData functions using the first 20 dimensions[77]. The most highly variable expressed genes were used for principal component analysis. The most significant principal components were used for UMAP dimension reduction. Cell clusters were determined by shared nearest-neighbor using the Louvain algorithm. Unbiased clustering was performed separately for each sample and for the combined samples to facilitate comparisons of subpopulations in HPV16+ and HPV16− rafts. Cluster gene lists (Supplementary Data 2) were determined by Seurat's FindAllMarkers functions using the Wilcoxon rank-sum test[77]. Genes that were uniquely expressed in each cluster were considered cluster-defining—these genes were expressed in the majority of cells in the cluster and minimally expressed (<30% of cells) or absent in all other clusters. A subset of these genes were selected for further analysis using quantitative RT-PCR and RNA-ISH. The scRNAseq data were deposited at the Gene Expression Omnibus (GEO) under accession number GSE189670.

## Pseudotime, including slingshot plot and individual gene pseudotimes

All cells were subjected to diffusion map dimension reduction and were labeled with the previously defined clusters from the UMAP plot. Monocle v2.22.0[25,78,79] was used to order cells in pseudotime. The original cell clusters from the Seurat pipeline were preserved to perform differential expression and subsequently order the cells. Analysis was performed with C9 cells included and excluded prior to ordering the cells. Cell cycle normalization was applied. The top 100 most significantly variable genes were plotted on a heatmap in pseudotime using scaled expression by row.

## Strategy for identifying cluster-defining genes

Detection of individual genes by RNA-ISH is a time- and cost-intensive assay. We, therefore, developed a strategy to narrow down candidate cluster-defining markers prior to RNA-ISH validation. Genes for RNA-ISH validation were selected based on (1) Expression in at least 80% of cells in the cluster of interest (percentage 1, pct.1) with minimal expression in remaining cells. (2) Robust cluster-specific expression in UMAP feature plots and in raft samples by RT-qPCR. (3) Availability of commercial RNA-ISH probes, and potential for combinatorial detection based on technical considerations, including channel availability. To this end, the gene list for each cluster was filtered by pct.1 ≥ 80% and sorted in increasing order by expression in the percent of remaining cells (percentage 2, pct.2) (Supplementary Data 3). The top 20 genes per cluster were considered; feature plots were visualized to confirm robust cluster-specific expression for each gene, RT-qPCR primers were designed for candidate genes, and expression was validated (CT <35) using RNA from monolayer cell extracts and subsequently raft extracts (Supplementary Data 3). Candidate cluster-defining genes were also evaluated by RT-qPCR for relative expression in HPV16− versus HPV16+ rafts and compared to expected relative ratios based on the scRNAseq data. Additionally, RNA expression was determined in subconfluent cells and overconfluent cells in 2D culture as well as in 3D rafts in order to determine the effect of differentiation and stratification. RNA-ISH validation was then performed in HPV+ and HPV− raft tissues for select cluster-defining genes in order to spatially localize clusters in 3D tissues. This validation of RNA-ISH probes for cluster mapping was performed in a minimum of three independently generated sets of HPV16+ and HPV16− rafts to ensure reproducibility.

## Analysis of published scRNAseq supplementary datasets

All individual samples from the published dataset[38] were combined and normalized individually by SCTransform v2[80]. Raw data are available on NCBI Sequence Read Archive: accession ID SRP301444. Processed gene barcodes are available on the Gene Expression Omnibus database: accession ID GSE164690. Samples were integrated based on anchor genes and identified by reciprocal PCA using the first 30

dimensions. Dimension reduction and clustering on the integrated Supplementary DataSet was performed as described above. Epithelial cells were further isolated by only using the samples that were from the CD45 negative samples and removing non-epithelial cell types, T-cells, endothelial, stromal, macrophage, and B-cells. Reclustering was performed as above.

## Ontology analysis
Functional enrichment analysis and conversions of gene lists was performed using the gProfiler web server (https://biit.cs.ut.ee/gprofiler/gost)[81]. The server includes Gene Ontology (GO) terms, curated by the Ensembl database[82], and pathways from KEGG (https://www.genome.jp/kegg/), WikiPathways (https://www.wikipathways.org/index.php/WikiPathways), and Reactome (https://reactome.org/). The Gene Ontology Statistics (GOSt) function was used to perform GO enrichment analysis of the input gene lists (cluster-defining genes/differentially expressed genes (DEGs)). GOSt uses multiple testing corrections by comparing the number of GO terms, KEGG pathways, etc., against the given input query. In addition, Benjamini–Hochberg FDR values for the GO terms were also included; FDR values less than 0.05 were considered significantly enriched (Supplementary Data 4).

## Cytoscape visualization
To visualize complex enriched gene sets, analysis was performed using Cytoscape, a software platform containing Apps for visualizing complex networks. To functionally characterize these complex Supplementary DataSets, the Enrichment Map[83] App within Cytoscape was used. In brief, enrichment files and gene sets outputs from g:Profiler were processed using Enrichment Map and terms with a false discovery rate less than 0.01 were considered significantly enriched. Nodes represent pathways, and they are connected by edges (lines) between nodes with genes in common. Node size is dependent on the enrichment score associated with that pathway, while edges thickness is relative to the number of genes shared between nodes.

## Transcription factor enrichment analyses
To identify transcription factors (TFs) that might play a role in the observed gene expression patterns, we employed complementary methods: TF binding site motif enrichment and ChIP-seq peak enrichment. For TF motif enrichment analyses, we used the HOMER software package[46] to calculate the enrichment of each motif in the promoter sequences (−200 bases to +50 bases relative to the gene TSS) of gene clusters with expression patterns of interest. HOMER was modified to use the large library of human position weight matrix (PWM) binding site models contained in build 2.0 of the CisBP database[84] and a log base 2 log likelihood scoring system. For ChIP-seq peak enrichment analysis, we used the RELI[47] algorithm. In brief, RELI calculates the enrichment of ChIP-seq peaks using a permutation-based method. RELI was run on the promoter sequences of gene sets of interest, using the "refGene_TSS_−1k_500" null model and a custom ChIP-seq library containing 11,504 human ChIP-seq datasets created by running MACS2[85] with default parameter settings on datasets obtained from the GEO database[86].

A final ranked list of TFs (Supplementary Data 6) was generated by combining the TF motif enrichment and ChIP-seq peak enrichment results. Prior to summarizing the HOMER data, results were preprocessed by expanding the TF components of each motif name into additional rows in order to account for multiple TFs recognizing a single motif. Unique TF entries were then summarized for HOMER and RELI datasets independently by selecting the record containing the lowest $P$ value and assigning a percentile score based on the total number of unique records in each dataset (HOMER $n = 1200$ TFs; RELI $n = 1487$ TFs). These ranked summaries were then merged into a common dataset using the name of the TF and subsequently filtered for records present in both HOMER and RELI ($n = 752$ TFs). Each TF was

then assigned a combined rank based on the average of the two percentile scores that were calculated in the respective summary datasets.

## Collection of HNSCC patient samples
Head and neck cancer specimens were collected at the time of planned surgical resection under an IRB-approved study protocol UCCI-UMB-14-01 (IRB #2014-4755), a general specimen collection protocol (IRB #2017-2137), and a CCHMC-approved protocol (IRB# 2009-2700). Samples were de-identified prior to transport to the laboratory. These studies were approved by the Institutional Review Board at the University of Cincinnati and Cincinnati Children's Hospital and were conducted in accordance with Good Clinical Practice guidelines and the Declaration of Helsinki. Written informed consent was received from all participating patients prior to enrollment. After surgical resection, tissue was placed in cold HypoThermosol FRS Preservation solution (Sigma-Aldrich, St Louis, MO, H4416) and transported directly to the laboratory on ice.

## Collection of cervical tissue with potential CIN lesions
De-identified cervical tissues were obtained from women who were undergoing loop electrosurgical procedures (LEEPs) or cold cone biopsies as part of clinical care based on the Risk-Based Management Consensus Guidelines released by the American Society for Colposcopy and Cervical Pathology. Cervical cones were transferred in conical tubes with media on ice to the University of Cincinnati Department of Pathology & Laboratory Medicine, where they were grossed per routine procedure. During processing, a small slice from each quadrant was removed, de-identified, and assigned a study ID before being transferred to research. Specimens were fixed, embedded, sectioned, H&E stained, and screened by certified pathologist review for areas of ectocervical tissue with HPV features or CIN lesions. This research activity was determined to be exempt from continuing IRB review by the University of Cincinnati (2021-0194) and Cincinnati Children's Hospital Medical Center (2021-0296).

## Generation of 3D organotypic rafts from the patient-derived cervical and tonsillar epithelium
Primary cervical keratinocytes from hysterectomy-derived ectocervix were kindly provided by Dr. Aloysius Klingelhutz[87]. Primary oral keratinocytes were isolated from tonsillectomy-derived palatine tonsils collected from Banner University Medical Center Tucson as approved by The University of Arizona Institutional Review Board[88]. Mycoplasma testing was performed regularly and cell lines were confirmed to be negative. Prior to rafting, primary cervical and tonsillar cells were maintained as monolayer cultures (as described above for NIKS) but with conditional immortalization using the Rho-kinase inhibitor Y-27632 (Chemdea, Ridgewood, NJ, CD0141) at 10 µM[89,90]. HPV18+ cervical cells were generated by removing Y-27632 from media and infecting with whole-genome HPV18 quasivirus at a multiplicity of infection of 100 viral genome equivalents per cell. Mock-infected cervical cells received an equivalent volume of only viral storage buffer (25 mM HEPES pH 7.5, 500 mM NaCl, 1 mM MgCl$_2$). HPV16+ tonsillar clones harboring episomal viral DNA were generated[91]. Cervical and tonsillar 3D organotypic rafts were both grown[88]. Briefly, rafts were constructed by creating a dermal equivalent, where hTERT-immortalized neonatal human foreskin fibroblasts (HFF-hTERT cells) were embedded ($8 \times 10^4$ per raft) into a solidified rat tail type I collagen matrix (Corning, Corning, NY, 354236). On the next day, HPV18 ± cervical or HPV16 ± tonsillar keratinocytes were seeded ($2.5 \times 10^5$ per raft) atop the dermal equivalent to create confluent basal layers. Finally, on the third day, cultures were lifted onto a hydrophilic polytetrafluoroethylene membrane insert (0.4-µm pore size; MilliporeSigma, Burlington, MA, PICM0RG50) for an air-liquid interface and fed every 2 days (seven total media changes) with differentiation media (1.88 mM

Ca$^{2+}$ and no epidermal growth factor, EGF) to stratify until fully differentiated raft cultures were harvested on day 15 post-lift.

## TCGA analysis

The TCGA PanCancer Atlas was analyzed using cBioPortal[92,93]. In the Head and Neck Squamous Cell Carcinoma Supplementary DataSet, 488 patients had complete data (including mRNA expression data), and were included in the analysis; 85% (415 patients) were classified as HPV- HNSCC, 15% (72 patients) as HPV+ HNSCC, and one patient was unknown. In the Cervical Squamous Cell Carcinoma Supplementary DataSet, 275 patients had complete data and were included in the analysis. ELF3 mRNA expression relative to all diploid samples with $z \geq 2.0$ was used to define the high ELF3 expression sub-group (10% of the HNSCC dataset and 9% of the cervical SCC dataset). A Pearson cutoff of $\geq |0.4|$ was applied to select genes correlating with ELF3 expression. Overlap between gene lists was evaluated using BioVenn[94] and confirmed manually.

## ELF3 knockdown experiments

Nontargeting and ELF3-specific short hairpin RNA (shRNA)-expressing lentiviral vectors were obtained through the Sigma MISSION shRNA program (Sigma-Aldrich, Inc., St Louis, MO, 68178). HPV16+ NIKS were transduced with NTsh or ELF3sh vectors (ELF3sh-1 TRCN0000013864; ELF3-sh2 TRCN0000013865, Sigma-Aldrich, Inc., St Louis, MO, 61878) and selected and maintained in media with 1 μg/mL puromycin.

## Western blot analysis

Antibody information, including dilutions and validations, are detailed in Supplementary Data 1. Organotypic raft lysates generated from ELF3 NT and ELF3 knockdown cells were prepared in RIPA buffer (10%SDS, 1 M Tris, 5 M NaCl, 0.25 M EDTA, Triton, sodium deoxycholate) containing HALT protease and phosphate inhibitor cocktail (Thermo Fisher Scientific, Waltham, MA,78443) and 200 μM Na$_3$O$_4$V. About 30 μg of protein was loaded per lane on 4–20% Bis-Tris gels (Bio-Rad, Hercules, CA, 5678094), and protein was transferred to Transblot turbo midi-size membrane (Bio-Rad, Hercules, CA, 1704275) at 200 mA for 2 h. Membranes were blocked in 10% dry milk reconstituted in TNET and incubated with ELF3 antibody (Sigma St Louis MO, HPA003479) at 1:1000 (Sigma St Louis MO, HPA003479-100ul) and Rhodamine Anti-actin (Bio-Rad, Hercules, CA, 12004163) at 1:5000 in TNET overnight at 4 °C. Membranes were incubated in ECL anti-Rabbit IgG secondary antibody (Thermo Fisher Scientific, Waltham, MA, NA934V) at 1:5000 in dry milk reconstituted.

## Statistics and reproducibility

scRNAseq cluster gene lists were determined by Seurat's FindAllMarkers functions using the Wilcoxon rank-sum test[77] as detailed above. Statistical analyses were performed using GraphPad Prism. Significance was set at $p < 0.05$ for all experiments, with the exception of the mutual exclusivity test on TCGA data, where significance was set at $p < 0.10$. For data with one independent variable, Mann–Whitney test with two-tailed distribution, unpaired $t$-test, or one-way ANOVA was used. For data with two independent variables, two-way ANOVA was used. The most appropriate tests for multiple comparisons following two-way ANOVA analyses was selected according to GraphPad recommendations and is specified per figure. Correlations were evaluated by Spearman's rank correlation test. Differences in Kaplan–Meier survival curves were tested by the log-rank test. Statistical values reported by cBioPortal on TCGA PanCancer Atlas data were noted. The number of replicates, statistical tests, and $p$ values are noted for each analysis in the figure legends.

## Reporting summary

Further information on research design is available in the Nature Portfolio Reporting Summary linked to this article.

## Data availability

The RT-qPCR data generated in this study are provided in the Source Data file. The scRNAseq data generated in this study have been deposited in the GEO database under accession code GSE189670. The analyzed scRNAseq data were available in Supplementary Information files. The scRNAseq publicly available data used in this study are available on the Gene Expression Omnibus database: accession ID GSE164690[38]. The TCGA data were accessed and are available through cBioPortal[92,93]. The remaining data were provided with this paper within the Article, Supplementary Information or Source Data file. Source data are provided with this paper.

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

## Acknowledgements

For the receipt of cervical samples, we thank the OB/GYN residents at the University of Cincinnati. We are thankful to Dr. Aloysius Klingelhutz for primary cervical keratinocytes, Dr. Deepta Bhattacharya and Dr. Lucas D'Souza for palatine tonsils, and Dr. Felicia Goodrum for HFF-hTERT cells to the Doorslaer laboratory. For assistance with the processing of clinical samples, we thank the pathology assistants, particularly Mojun Zhao, in the Department of Pathology at the University of Cincinnati. For reagents and discussions, we thank all members of the Wells laboratory. We thank Sarah McLeod and Evan Meyer in the CCHMC Confocal Imaging Core for hands-on support. This study was supported by services from the Pathology Research Core shared facility in the Cincinnati Children's Research Foundation with excellent technical expertize provided by Meredith Taylor and Lynn Duncan. Histology services were additionally provided by the University of Arizona Cancer Center's Tissue Acquisition and Cellular/Molecular Analysis (TACMASR) shared resource core which is supported by the National Cancer Institute of the National Institutes of Health (NIH) under award number P30 CA023074. This research was funded by the following NIH grants: T32ES007250 (M.C.B.), F30AI157229 (M.C.B.), CA228113 (S.I.W.), U01AI130830 (M.T.W.), R01HG010730 (M.T.W.), R03DE030211 (K.V.D.), as well as support from CancerFreeKids (M.C.B.), the State of Arizona Improving Health TRIF (K.V.D), the BIO5 Institute (R.J.), and the Natural Sciences and Engineering Research Council of Canada PDF-546182-2020 (R.J.)

## Author contributions

M.C.B., M.G.B., M.A., S.S.P., and S.I.W. conceptualized and designed the project. M.C.B, T.C., A.C., M.G.B., D.L., J.C., G.N., R.J., and D.E.J.W. conducted experiments and analyzed the data. M.C.B., A.V., M.A., M.T.W., B.A., and P.D. performed bioinformatic analyses. M.R., M.K., M.E.R., A.K., R.L.F., and K.V.D. provided experimental training and conceptual input. M.L., T.M.W.-D., P.F.L., R.J., D.E.J.W., and K.V.D. provided key reagents, samples, and/or protocols. M.C.B. coordinated cervical tissue collections supervised by M.C., A.J., C.B., and T.J.H., and HNSCC tissue collections supervised by A.T., C.Z., Y.P., M.L., and T.M.W.-D. A.C. assisted in the transport and processing of tissues. G.N., A.K., and K.A.W.-B. provided board-certified pathology analyses. M.T.W., P.F.L., M.A., S.S.P., K.V.D., and S.I.W. supervised the project. M.C.B. and T.C. wrote the manuscript, with edits and revisions by M.C.B., K.A.W-B., P.F.L., M.A., and S.I.W. All authors provided important intellectual contributions and critical feedback and approved the manuscript.

## Competing interests

R.L.F: Adagene Incorporated: Consulting; Aduro Biotech, Inc: Consulting, AstraZeneca/MedImmune: Clinical Trial, Research Funding; Bicara Therapeutics, Inc: Consultant; Bristol-Myers Squibb: Advisory Board, Clinical Trial, Research Funding; Brooklyn Immunotherapeutics LLC: Consultant; Catenion: Consultant; Coherus BioSciences, Inc.: Advisory Board; Eisai Europe Limited: Advisory Board; EMD Serono: Consultant; Everest Clinical Research Corporation: Consultant; F. Hoffmann-La

Roche Ltd: Consultant; Federation Bio, Inc: Consultant; Genocea Biosciences, Inc: Consultant; Genmab: Advisory Board; Hookipa Biotech GmbH: Advisory Board; Instil Bio, Inc: Advisory Board; Kowa Research Institute, Inc.: Consultant; Lifescience Dynamics Limited: Advisory Board; MacroGenics, Inc.: Advisory Board; MeiraGTx, LLC: Advisory Board; Merck: Advisory Board, Clinical Trial; Mirati Therapeutics, Inc: Consultant; Mirror Biologics Inc: Supplementary Data Safety Monitoring Board; Nanobiotix: Consultant; Novartis Pharmaceutical Corporation: Consulting; Novasenta: Consulting, Stock, Research Funding; Numab Therapeutics AG: Advisory Board; OncoCyte Corporation: Advisory Board; Pfizer: Advisory Board; PPD Development, L.P.: Consultant; Rakuten Medical, Inc: Advisory Board; Sanofi: Consultant; Seagen, Inc: Advisory Board; SIRPant Immunotherapeutics, Inc: Advisory Board; Tesaro: Research Funding; Vir Biotechnology, Inc: Advisory Board; Zymeworks, Inc.: Consultant. T.J.H.: AstraZeneca, Caris, Clovis, Eisai, Genentech, GlaxoSmithKline, Immunogen, Johnson & Johnson, Merck, Mersana and Seagen (Advisory Boards). The remaining authors declare no competing interests.

## Additional information

[1]Division of Oncology, Cincinnati Children's Hospital Medical Center, Cincinnati, OH 45229, USA. [2]School of Animal and Comparative Biomedical Sciences, University of Arizona, Tucson, AZ 85721, USA. [3]Division of Clinical Operations, Medpace, Cincinnati, OH 45227, USA. [4]McArdle Laboratory for Cancer Research, School of Medicine and Public Health, University of Wisconsin, Madison, WI 53705, USA. [5]Center for Autoimmune Genomics and Etiology (CAGE), Cincinnati Children's Hospital Medical Center, Cincinnati, OH 45229, USA. [6]Division of Allergy and Immunology, Cincinnati Children's Hospital Medical Center, Cincinnati, OH 45229, USA. [7]Division of Biomedical Informatics, Cincinnati Children's Hospital Medical Center, Cincinnati, OH 45229, USA. [8]William G. Lowrie Department of Chemical and Biomolecular Engineering, Ohio State University, 151 W. Woodruff Ave, Columbus, OH 43210, USA. [9]Department of Pathology, Ohio State University Medical Center, Columbus, OH 43210, USA. [10]Division of Hematology/Oncology, Department of Internal Medicine, University of Cincinnati College of Medicine, Cincinnati, OH 45267, USA. [11]Cancer Biology Graduate Interdisciplinary Program, University of Arizona, Tucson, AZ 85721, USA. [12]Medical Scientist Training M.D.-Ph.D. Program (MSTP), College of Medicine-Tucson, University of Arizona, Tucson, AZ, USA. [13]UPMC Hillman Cancer Center, University of Pittsburgh, Pittsburgh, PA 15232, USA. [14]Tumor Microenvironment Center, UPMC Hillman Cancer Center, Pittsburgh, PA 15232, USA. [15]Department of Obstetrics and Gynecology, University of Cincinnati Medical Center, Cincinnati, OH 45267, USA. [16]Department of Otolaryngology, University of Cincinnati, Cincinnati, OH 45267, USA. [17]Department of Otolaryngology, University of Pittsburgh, Pittsburgh, PA, USA. [18]Department of Immunology, University of Pittsburgh, Pittsburgh, PA 15232, USA. [19]Department of Pathology & Laboratory Medicine, University of Cincinnati College of Medicine, Cincinnati, OH 45267, USA. [20]Division of Developmental Biology, Cincinnati Children's Hospital Medical Center, Cincinnati, OH 45229, USA. [21]Department of Pediatrics, University of Cincinnati College of Medicine, Cincinnati, OH 45267, USA. [22]Divisions of Human Genetics, Biomedical Informatics and Developmental Biology, Cincinnati Children's Hospital Medical Center, Cincinnati, OH 45229, USA. [23]The BIO5 Institute, University of Arizona, Tucson, AZ 85721, USA. [24]Department of Immunobiology, University of Arizona, Tucson, AZ 85721, USA. [25]UA Cancer Center, University of Arizona, Tucson, AZ 85721, USA. [26]Division of Pathology & Laboratory Medicine and The Perinatal Institute Division of Pulmonary Biology, Cincinnati Children's Hospital Medical Center, Cincinnati, OH 45229, USA. [27]These authors contributed equally: Mary C. Bedard, Tafadzwa Chihanga. ✉e-mail: Mike.Adam@cchmc.org; Steve.Potter@cchmc.org; Susanne.Wells@cchmc.org

