## [Peer Review File · Nature Communications]

Single cell transcriptomic analysis of HPV16-infected epithelium identifies a keratinocyte subpopulation implicated in cancerReviewers' Comments:

Reviewer #1:

Remarks to the Author:

This study by Bedard et al., provides, to my knowledge, the first characterization of transcriptomics in HPV infected epithelium as compared to an uninfected control epithelium derived from an isogenic cell background. This is important information particularly since HPV16 causes $\approx 50\%$ of cervical cancers and $\approx 90\%$ of head and neck SCCs. Noteworthy findings include the compelling finding of a unique keratinocyte subpopulation (C8) wherein HPV16 uses endogenous cellular signaling pathways to simultaneously maintain cellular proliferation and differentiation critical for viral survival and propagation. Another predominantly HPV+ subpopulation (C9) was characterized more in-depth, but jumped from a productive HPV16 lesion phenotype into SCC, which was not well rationalized.

A strength of the work is the comparison of transcriptomics between HPV+ and HPV- epithelium from an isogenic background. The authors note many positive aspects of this model in the Discussion, which appears accurate. However, the use of the previously immortalized NIKS cells (from male foreskin keratinocytes) as the background cell also has a number of limitations that may obscure the understanding of how HPV alters gene expression. The authors should also point out some of the main limitations of this model using previously immortalized cells in understanding HPV biology.

The most noteworthy weakness/limitation of this work is that it is not clear how many independent HPV- and HPV16+ raft cultures were separately analyzed by scRNAseq. If only one of each, or pooled samples of each, the data are of unclear significance, rigor, and reproducibility. I could not find any notation of the total number of cells that were analyzed for each sample. These issues are potentially major weaknesses. Another possible weakness is the use of normal skin (from an undefined location) as a comparator to p16+ cervical lesions. Since HPV16 does not cause overt pathology outside of the oral or genital mucosa, and the basis for this is not defined, the value of normal skin markers is not clear.

The manuscript (particularly the Results section) is quite dense and lengthy, and there are concerns that it will not be readily understandable by a wide audience since it was difficult to follow based on HPV biology alone. There are other scRNAseq analyses of epithelium published that appear more readable. On the other hand, the Discussion was very short and could better have highlighted major findings. I wonder if this study could be separated into two discrete publications to better enable the digestion of the huge amount of data. The value of including data from subconfluent and overconfluent HPV-negative cells is not clear and seems to detract from the overall goal of the manuscript. Further, the definition of "subconfluent" and "overconfluent" was not clear. The following sentence in the Discussion is difficult to understand "While the role of the C9 population in human SCC development is not yet clear, the location of proliferating HPV16+ basal cells in association with their differentiated, nonproliferative C9 progeny indicates supportive activities of the latter for the former." The choice of genes and proteins to test for spatial expression in epithelial tissues was not well rationalized.

A noteworthy finding was a so-called "differentiation-dissonant keratinocyte population" in HPV16+ rafts that appears to require the transcription factor ELF3 (the C9 population). A caveat to this finding is that, according to the Human Protein Atlas (www.proteinatlas.org), ELF3 and LCN2, but not GPR110, are normally expressed in the upper layers of the cervix. This differs from the IF data on the authors' raft tissues, where HPV16+, but not HPV- NIKS tissues, express ELF3, LCN2, and GPR110 in the upper layers. This may indicate additional limitations of using the immortalized NIKS background cells and further suggests the HPV16+ raft tissues have a more normal phenotype with regard to these factors as compared to the HPV- NIKS "control" rafts. These issues should be discussed in the manuscript.

A concept difficult to rectify is the association between increased markers only present in the upper

differentiated epithelial layers (e.g., ELF3) and how they might contribute to cancer, which by definition, must penetrate the basement membrane. Since the uppermost layers of the HPV16+ raft epithelium both promote late virus activities as a productive lesion and would (at least in vivo) be destined to be sloughed off and lost, how do the authors envision these cells could contribute to cancer? It might have been helpful to know how the expression of these markers changes in a cohort (rather than a couple of HNSCC and CIN2 tissues) during CIN progression to CIS. To be clear, the HPV16 raft model is more akin to a benign productive lesion, and it is a bit premature (or at least confusing) to jump with analyses into SCC. Perhaps the cancer analyses are better fitted to a separate manuscript; keeping the focus of this manuscript focused on C8 and C9 phenotypes in early neoplasia may make the story less dense and easier to digest. Again, it just seems like a large jump from the role of the C9 function in productive HPV infection to cancer.

There are misreferenced figures in the text (and the authors should verify all):

Page 11, columns 1-2: Figs. noted as Fig. 5, should be Fig. 8. Perhaps also on page 13?

Page 11, column 2: "Additionally, expression of C9 markers was increased in models of differentiation (overconfluent 2D cultures and 3D rafts), relative to undifferentiated (subconfluent) 2D cultures (Fig. 1C)." Fig. 1C should be Fig. S1C.

Could not find Fig. S4D and 6 (noted on page 13) or Fig. 6K, 6L (should be 9K and 9L?) noted in the legend to Fig. S1

Fig. S1 seems to use different nomenclature for genes in the legend versus some of the figures (MACC1, CLDN4), which is confusing.

Page 14, column 1 mentions Fig. 7B, but this does not seem to be accurate.

Reference 23 seems to be for reovirus and not HPV.

It would be helpful if the colors of blue for downregulated genes and yellow for upregulated genes in the heat maps were maintained in all cytoscape plots and bar graphs for consistency. (e.g., Fig. 5 C,D seems inverse).

For example, is abbreviated "e.g.", not "eg".

Reviewer #2:

Remarks to the Author:

Bedard et al. applied scRNA-seq to NIKS (normal immortalized keratinocytes), with and without HPV infection, grown in organotypic 3D rafts, in order to assess the effects of HPV on infected epithelial tissue at single cell resolution. Some of the findings were further validated through RNA-ISH of sections from other NIKS as well as patient samples. The analysis revealed 12 clusters of epithelial cells, with 11 clusters that are shared between HPV+ and HPV- cells and only one that is unique to HPV+ cells. However, multiple clusters are biased towards either HPV+ or HPV- cells. These differences are primarily related to the cell cycle, which is increased, as expected, in HPV+ cells and to differentiation processes. The authors focus on cluster 9 (C9) that is the most distinct from all other clusters, is enriched in (but not specific to) HPV+ cells, has a specific spatial localization to the uppermost layer of HPV-infected rafts, and is characterized by ELF3 mRNA expression. It is then demonstrated that C9 cells are also present in HPV+ cancer tissues, and that ELF3 expression is correlated with prognosis of cervical cancer and that it regulates the C9 cells.

Given the significance of HPV infection and the limited knowledge of HPV effects at the tissue level, the general direction of this study is important and the experimental setup is interesting although not entirely cutting edge (as much more advanced spatial technologies have recently become widely available). The experiments appear to be executed at a high level. The analysis is performed at a medium level, which suggests only partial understanding of the methods and suboptimal flow and presentation of the results. The identification of C9 is of interest, although the origin, function and significance of this population remain quite vague. This population is not unique to HPV+ cells, its

association with survival does not suggest causation and its biology was only described in a preliminary manner. Overall, the study is of significance but lacks a major finding or conceptual advance.

Major comments

1. The manuscript seems long and non-focused with suboptimal flow and too many figures. Accordingly, I found it difficult to read the manuscript and connect the different pieces into a coherent picture. In my opinion, the authors should identify the main and most reliable results that are of broad interest, focus on them while improving the flow and the presentation of those results, and move the remaining analyses to the supplement. A shorter and more focused paper could be considerably improved.

2. Pseudotime analysis should be used and interpreted with caution and it is not suitable for connecting well-separated clusters that do not lie on a continuous trajectory. In this particular case, C9 seems completely distinct from the other clusters and is "forced" onto the trajectory. Its inclusion in the pseudotime analysis is problematic and the results are not reliable.

3. To demonstrate the relevance of C9 for in vivo HPV+ disease the authors analyzed normal and disease tissues by IF and ISH with markers of C9. However, the IF results cannot be directly compared to the ISH results. Such comparative analyses are only relevant if they rely on exactly the same protocol.

4. The authors argue that the association of C9 with survival cannot be analyzed in HNSCC due to variation between sites and limited sample numbers. The first issue can be avoided by analyzing each site separately and focusing on the sites with the highest statistical power. The second issue can be approached by presenting the results despite their limited statistical power. Even if the effect is not significant I would argue that the data should be shown and interpreted in light of the limited statistics. For example, a non-significant trend towards C9 correlating with poor survival would still be partially consistent with such association, while a complete lack of correlation or a non-significant opposite trend (towards the correlation in the opposite direction) may not be consistent with such association even with limited statistical power.

5. As I understand the experimental setup, HPV-infected and non-infected NIKS were sequenced together and the NIK of origin was determined by HPV16 mRNA expression. While the authors do show ubiquitous HPV16 DNA expression this does not necessarily translate to HPV reads being identifiable when performing scRNAseq, especially with the known problem of dropouts. Thus, when the authors compare HPV+ and HPV- cells between different clusters, some apparent HPV- cells could actually originate from the HPV-infected NIK, and some of the clustering could be due to batch effects.

6. Did any of the HPV-infected cells undergo further carcinogenic transformation? This may be explored by pathology analysis, inferring copy number aberrations etc.

7. In figure 5A ribosomal genes are shown to be enriched in HPV- cells. In scRNAseq, ribosomal genes often indicate a quality issue rather than actual biology: since they are the most highly expressed genes in cells, they are often affected by the global quality and coverage of single cell data and may appear as upregulated or downregulated due to technical reasons. The fact that figure 8H shows mitochondrial genes, a proxy for dying cells, to be upregulated in HPV- strengthens this point, as does the generous QC threshold of 30% mitochondrial genes per cell. The authors should show that QC parameters are comparable between HPV- and HPV+ cells and ensure that the results are robust to the use of more strict QC filtering.

Minor comments

1. It may be helpful to name the clusters with informative names based on their top marker genes.
2. Functional studies of the impact of ELF3 overexpression/depletion on tissue phenotype are not

necessary but could improve the study.

3. In figure 2F, HPV+ cells are said to be enriched in certain clusters. This should be done stringently with a statistical test and p-value shown.

4. Figure 4B and other similar figures show two repeats with no significance values. More repeats and statistical testing should be added.

5. P-values and statistical tests used for GO enrichment analysis should be shown.

6. If no C8-specific markers were found, what makes this a distinct population?

7. On page 11, references to figure 5 actually refer to figure 8.

Reviewer #3:

Remarks to the Author:

In this manuscript, the authors aimed to explore the keratinocyte subpopulations which support distinct phases of the viral life in persistent HPV16 infection. To this purpose, the authors generated a HPV16-isogenic organotypic epithelial raft model as it mimics the effects of HPV infection on epithelial biology. By scRNAseq to generate a hostpathogen transcriptome atlas of HPV16+ versus HPV16- keratinocytes, an HPV-reprogrammed keratinocyte subpopulation (C9) has been identified in a new surface compartment of the squamous epithelium. The author found that this subpopulation required overexpression of the ELF3/ESE-1 transcription. Moreover, TCGA analyses of patient tissues demonstrated that high ELF3 expression levels were associated with HPV in premalignant and malignant disease. The final ELF3 knockdown experiments suggest that ELF3 is required for formation of the C9 compartment at the surface of stratified HPV16+ epithelium. The authors conclude that ELF3 might be a potential therapeutic target in the fight against HPV persistence and carcinogenesis.

The combined use of HPV-infected 3D culture technique and scRNAseq is very interesting. In my opinion the association between the new single cell technology and the biological approach has generated a significant biological insight into the cell populations under study for such a high-impact general interest journal. The analyses and methods performed are complete and consistent with answering the specific question. The results are robust and fit-for-purpose and the controls and sampling mechanisms are sufficient and well described.

I recommend this manuscript for publication with no revisions.

Response to reviewer comments. We appreciate detailed comments by the reviewers, and have provided responses to each below. Corresponding changes in the revised manuscript are indicated in blue. To focus the manuscript, validate C9 broadly, and rationalize inclusion of C9 in HPV+ premalignancies and malignancies, the following 37 subfigures were newly added and 12 subfigures were revised in response to reviewers' feedback. Subfigures that were added include: Fig. 3A-D, Fig. 4F-G, Fig. 5D-I, Fig. 6A-F, Fig. 7A-F, Table 1, Fig. 9A-E, Fig10. A-F, H, and Fig. 11A. Subfigures that were revised include: Fig. 4A, C, D, Fig. 5 A-E, Fig. 6A, B, Fig. 11G, and Fig. S2C. A total of 26 subfigures from the previous manuscript that convoluted the focus of the manuscript away from the main C9 story were removed to be published elsewhere as recommended. We believe these changes have substantially improved the focus and flow of the manuscript, and that we have addressed reviewer concerns. We have indicated our changes from the original manuscript to the revised manuscript with blue text.

Reviewer #1, expertise in HPV carcinogenesis (Remarks to the Author):

1. This study by Bedard et al., provides, to my knowledge, the first characterization of transcriptomics in HPV infected epithelium as compared to an uninfected control epithelium derived from an isogenic cell background. This is important information particularly since HPV16 causes ≈50% of cervical cancers and ≈90% of head and neck SCCs. Noteworthy findings include the compelling finding of a unique keratinocyte subpopulation (C8) wherein HPV16 uses endogenous cellular signaling pathways to simultaneously maintain cellular proliferation and differentiation critical for viral survival and propagation. Another predominantly HPV+ subpopulation (C9) was characterized more in-depth, but jumped from a productive HPV16 lesion phenotype into SCC, which was not well rationalized.

We thank reviewer 1 (R1) for their appreciation of HPV as a major cause of cancer, highlighting the importance of this work as the first report of a single cell transcriptomics map in HPV infected epithelium, and the significance of our choice to generate this novel HPV atlas in an isogenic model. We agree that our focused study of C9, now referred to "HPV-induced differentiation-dissonant epithelial nonconventional" (HIDDEN) cells, in malignant lesions was not well rationalized in our initial submission. This oversight was also noted by Reviewer 2. Our rationale is now described on page 11, and supported by the new figures 7A-F, 9A-E, and 10A-H to demonstrate that HPV-driven HIDDEN cell enrichment is broadly observed in high-risk HPV-infected as well as in HPV+ premalignant and malignant lesions, giving additional support for clinical significance of this subpopulation in HPV driven disease. Specifically, the HPV-induced HIDDEN cell population is present in HPV16+ primary tonsillar and HPV18+ primary cervical rafts as compared to the respective uninfected counterparts, in an expanded panel of HPV+ CIN lesions versus control tissues, and in HPV-positive as compared to HPV-negative head and neck cancers. Additionally, HIDDEN cell detection in head and neck cancer specimens was complemented with recently published single cell sequencing data from HPV+ and HPV- tumors (new Fig9). Integration of this independently derived dataset published by the Ferris laboratory validated the intriguing enrichment of HIDDEN cells in HPV+ tumors. This significant advancement of our studies enabled by the collaborative support of Drs. Ferris, Kulkarni, Van Doorslaer, Williams and Jackson is appropriately acknowledged by adding them as co-authors on this revised manuscript.

2. A strength of the work is the comparison of transcriptomics between HPV+ and HPV- epithelium from an isogenic background. The authors note many positive aspects of this model in the Discussion, which appears accurate. However, the use of the previously immortalized NIKS cells (from male foreskin keratinocytes) as the background cell also has a number of limitations that may obscure the understanding of how HPV alters gene expression. The authors should also point out some of the main limitations of this model using previously immortalized cells in understanding HPV biology.

The reviewer points out an issue in the HPV field wherein the many distinct sites of HPV infection in the human body are not reflected by a wide variety of cellular models. Although NIKS are a classical model for studies of HPV, important limitations include an extra isochromosome of the long arm of chromosome 8, the relative absence of productive viral amplification which was reflected in our scRNAseq data, the immortalized nature of the NIKS, and their male origin from human foreskin. While high risk HPV associated penile cancer can originate in foreskin, the incidence of penile cancer is rare when compared to cervical and head and neck cancer. As suggested by the reviewer, these main limitations of using immortalized NIKS as a model to understand HPV biology and additional limitations of this model system are now detailed in the Discussion (page 17). To directly address these limitations, we now demonstrate that high-risk HPV16 and HPV18 also induce the HIDDEN C9

cellular compartment in tonsillar and cervical epithelial rafts, respectively (Fig. 7A-D, Fig. S4A-C and Fig 10A-F). These newly added experiments extend our findings in NIKS rafts to primary cervical and head and neck mucosal rafts enhancing the clinical relevance of our discoveries for HPV induced cervical and head and neck carcinogenesis.

3. The most noteworthy weakness/limitation of this work is that it is not clear how many independent HPV- and HPV16+ raft cultures were separately analyzed by scRNAseq. If only one of each, or pooled samples of each, the data are of unclear significance, rigor, and reproducibility. I could not find any notation of the total number of cells that were analyzed for each sample. These issues are potentially major weaknesses. Another possible weakness is the use of normal skin (from an undefined location) as a comparator to p16+ cervical lesions. Since HPV16 does not cause overt pathology outside of the oral or genital mucosa, and the basis for this is not defined, the value of normal skin markers is not clear.

The numbers of sequenced cells are now clearly indicated in the materials and methods section (page 25). One isogenic HPV16+/HPV- raft culture scRNAseq experiment was carried out that is now clearly stated on page 5, and associated caveats are discussed. Our rationale for performing the initial labor and cost intensive scRNAseq discovery studies in isogenic HPV+ and HPV- immortalized NIKS rafts was to capitalize on the benefits of using this controlled isogenic model system to uncover novel HPV-induced cell populations that would be more challenging to uncover in primary patient tissues and models which inherently have more biologic variation. This initial discovery step led to identification of novel HPV induced subpopulations that we then chose to interrogate in multiple HPV induced human diseases using other raft models and primary patient samples to optimize the clinical relevance of our novel cell populations uncovered by scRNAseq analysis of the HPV+/HPV- NIKS isogenic model. Additionally, since the ultimate goal of the studies was to generate a physical atlas of keratinocyte subpopulations, we chose to place our emphasis on validating and geographically localizing newly identified HPV-induced squamous epithelial sub-populations in three independently generated isogenic HPV16+/HPV- NIKS organotypic rafts and in rafts generated from human primary tonsillar and cervical mucosal rafts to extend our discoveries into clinically relevant disease samples. This was not explained well, nor shown in the original version of the manuscript. We have now added, in this revised manuscript detailed analyses of three independently generated isogenic sets of rafts, and quantified total RNA puncta/ protein and geography of KLK7+ terminally differentiated (C3) and HIDDEN (C9) cells relative to each other in HPV16+ versus HPV- rafts using cell type marker detection in tissues by RNA in situ hybridization and immunofluorescence (Fig4F-G, 5D-F, 5H-I, and S2D-E). These quantitative studies now demonstrate that the HIDDEN but not the KLK7+ cell subpopulation is enriched in HPV+ stratified squamous epithelium derived from cutaneous, cervical, and tonsillar tissues as suggested by the single cell transcriptome. Furthermore, our quantitative geographic studies in the rafts clearly demonstrate a shift from the KLK7+ subpopulation being the prominent cell type within the superficial epithelial layers in uninfected epithelium (Fig4F-G) to the HIDDEN subpopulation instead being the prominent cell type within the superficial epithelial layers of the HPV-infected epithelium (Fig5D-F, H-I).

We agree that normal skin is a suboptimal negative control for our studies, with HPV negative HNSCCs and non-diseased cervical mucosal tissues representing better biological and clinically relevant controls. Thus, the human skin control used to demonstrate C9 absence in human epidermis as control for the findings in HNSCC biopsies was removed. Low baseline expression of the HIDDEN C9 subpopulation via ELF3 detection is now demonstrated in non-diseased tonsillar tissue as an optimal tissue of origin control for the HPV+ HNSCC studies, (Fig 9A). Non-diseased cervical tissue confirmed by a board-certified pathologist is now used as control tissue for studies involving the CIN studies to simultaneously detect the HIDDEN C9 subpopulation and HPV associated p16 protein expression (Fig7E). Additionally, we have shown upregulation of HIDDEN cells in rafts derived from more clinically relevant tissue types, including cervical and tonsillar epithelium, to allow direct comparisons between HPV+/- tissues.

4. The manuscript (particularly the Results section) is quite dense and lengthy, and there are concerns that it will not be readily understandable by a wide audience since it was difficult to follow based on HPV biology alone. There are other scRNAseq analyses of epithelium published that appear more readable. On the other hand, the Discussion was very short and could better have highlighted major findings. I wonder if this study could be separated into two discrete publications to better enable the digestion of the huge amount of data.

We thank the Reviewer for the suggestion to consider separating the manuscript into two separate manuscripts given the expansiveness of the data and lengthy results section. Thus, we have focused this manuscript on the HIDDEN and KLK7+ subpopulations enabling a more concise, understandable results section complemented by an expanded Discussion of the major findings together with the biologic and clinical relevance. To this end, we have extensively revised the results section with three entirely new main figures (Fig. 7, 9, 10), and added many new sub-figures to previous figures to include KLK7+/C3 validation (Fig4F-G, previously Fig3) and HIDDEN/C9 validation (Fig5D,E,F,H,I and Fig S2D,E; previously Fig6 and FigS4). The previous Figures 4,5, 7 and 8 and supplementary figures Fig S2 and Fig S3 were removed for a separate manuscript as recommended by Reviewers 1 and 2 to focus the manuscript and enhance the understanding of the primary discoveries by a broader audience.

5. The choice of genes and proteins to test for spatial expression in epithelial tissues was not well rationalized...The value of including data from subconfluent and over-confluent HPV-negative cells is not clear and seems to detract from the overall goal of the manuscript. Further, the definition of “subconfluent” and “overconfluent” was not clear...

Our rationale for selecting genes and proteins to test for spatial expression is now clearly outlined in the manuscript on page 6. In short, we selected candidate cluster-defining genes that satisfied three criteria: **1)** Expression in at least 80% of cells in the cluster of interest. **2)** Robust expression in rafts by RT-qPCR (CT<35) that was limited to the cluster of interest in UMAP feature plot visualization. **3)** Availability of commercial RNA-ISH probes and potential for combinatorial detection based on channel availability.

Determining the differential expression of cluster-defining markers in undifferentiated sub-confluent and differentiating super-confluent cells was critical for identifying HIDDEN cells as these studies demonstrated that development of the HIDDEN cell phenotype required a differentiation-inducing environment. Specifically, HIDDEN cells did not express most conventional differentiation markers but were only detected in differentiated over-confluent cells and rafts. This demonstrated that while the HIDDEN cell subpopulation is programmed for a unique differentiation program that differs from usual stratified squamous epithelial differentiation (termed differentiation dissonant), formation of the HIDDEN cell phenotype required a differentiation inducing environment providing mechanistic insights into why this subpopulation is localized to the superficial epithelial layers within rafts. Our data provide evidence that models incorporating differentiation are required for identifying and investigating HIDDEN cells and an explanation for why this transcriptionally unique population has eluded identification in the vast majority of HPV related studies carried out in traditional monolayer culture models. Based upon this rationale, we have chosen to retain these studies in the manuscript wherein we identify and phenotype this novel cell population and integrate it with the existing literature in the field of HPV biology. To clarify the goals of the experiments, the ‘subconfluency’ and ‘overconfluency’ terms are now defined in the materials and methods (page 24). Additionally, further Discussion on this topic is provided on pages 8-9 and 17 to ensure that the data contribute to rather than distract from the overall goals of the manuscript.

6. The following sentence in the Discussion is difficult to understand “While the role of the C9 population in human SCC development is not yet clear, the location of proliferating HPV16+ basal cells in association with their differentiated, nonproliferative C9 progeny indicates supportive activities of the latter for the former.”

We thank the reviewer for identifying this confusing sentence. We have refined the statement in the Discussion on pages 17-18: “Multiple scenarios could be at play for a hypothesized role for HIDDEN cells in worsening outcomes for HPV+ cancers. HIDDEN cell effects on basal cells might occur via paracrine activities that create a pro-tumor microenvironment, and particularly in severely dysplastic lesions where ELF3+ HIDDEN cells appear to expand downwards into the basal layer (Fig. 7F). Such activities may act on aspiring tumor cells to promote proliferation or survival, or may reprogram the tumor microenvironment to promote angiogenesis or suppress immune surveillance. Alternatively, the HIDDEN keratinocyte signature harbored non-conventional differentiation markers including select mucins and enzymes involved in o-linked glycosylation. HIDDEN cells might therefore provide a physical barrier against exogenous anti-viral or anti-tumor responses. Lastly, HIDDEN cells featured gene markers characteristic of primitive stem/progenitor cells such as KLF5 and K15. It is therefore also possible that these cells harbor quiescent stem-like features which might convert to a proliferative state during tumor evolution. Such potential roles remain to be identified.”

7. A noteworthy finding was a so-called “differentiation-dissonant keratinocyte population” in HPV16+ rafts that appears to require the transcription factor ELF3 (the C9 population). A caveat to this finding is that, according to the Human Protein Atlas (www.proteinatlas.org), ELF3 and LCN2, but not GPR110, are normally expressed in the upper layers of the cervix. This differs from the IF data on the authors’ raft tissues, where HPV16+, but not HPV- NIKS tissues, express ELF3, LCN2, and GPR110 in the upper layers. This may indicate additional limitations of using the immortalized NIKS background cells and further suggests the HPV16+ raft tissues have a more normal phenotype with regard to these factors as compared to the HPV- NIKS “control” rafts. These issues should be discussed in the manuscript.

We appreciate this important comment. We did not intend to imply that cells positive for markers of HIDDEN cells are a new cell population exclusively observed in HPV+ tissues, but rather that they are induced by HPV. The acronym HIDDEN reinforces this idea and we have now stated clearly that HIDDEN cells are present in HPV- tissues. The additional quantifications in Fig5D-F and H-I validate that while HIDDEN cells are detectable in uninfected NIKS rafts, they are greatly enriched in HPV16-infected rafts where they form a unique surface compartment.

Next, rather than solely rely on the Human Protein Atlas to determine relative expression, we have included additional controls in our experiments. In the protein atlas, ELF3 and LCN2 RNA expression was indeed reported for the cervix. However, based on protein expression, both proteins were only detectable in glandular, as opposed to squamous epithelial cells that are the focus of this work. The newly added uninfected organotypic rafts generated from primary cervical cells (Fig7A) and a normal cervical specimen (Fig7E) as controls for HPV+ primary cervical rafts and HPV+ CIN lesions are much more powerful in directly assessing relative expression at the RNA and protein level. Similarly, the addition of a normal tonsillar specimen (Fig 10A-B) and derivative tonsillar rafts (Fig. 10C) are more authentic controls for HPV+ epithelium and HNSCCs.

Finally, as described above, we have addressed the limitations of NIKS rafts by directly detecting the ELF3+ compartment in HPV16+ tonsillar and in HPV18+ cervical rafts. Given these extensive validation experiments, we are confident that HPV- NIKS rafts are a representative model of normal squamous epithelium in the context of HIDDEN cells, albeit with caveats to consider.

8. A concept difficult to rectify is the association between increased markers only present in the upper differentiated epithelial layers (e.g., ELF3) and how they might contribute to cancer, which by definition, must penetrate the basement membrane. Since the uppermost layers of the HPV16+ raft epithelium both promote late virus activities as a productive lesion and would (at least in vivo) be destined to be sloughed off and lost, how do the authors envision these cells could contribute to cancer? It might have been helpful to know how the expression of these markers changes in a cohort (rather than a couple of HNSCC and CIN2 tissues) during CIN progression to CIS. To be clear, the HPV16 raft model is more akin to a benign productive lesion, and it is a bit premature (or at least confusing) to jump with analyses into SCC. Perhaps the cancer analyses are better fitted to a separate manuscript; keeping the focus of this manuscript focused on C8 and C9 phenotypes in early neoplasia may make the story less dense and easier to digest. Again, it just seems like a large jump from the role of the C9 function in productive HPV infection to cancer.

A multitude of new data has focused our revised manuscript on the detection and regulation (by ELF3) of the HIDDEN cell subpopulation. Characterizing the hypothesized pro-carcinogenic role of C9 via functional studies will require extensive additional work that is beyond the scope of this manuscript particularly since it can only be studied in differentiated models. Such functions may differ in the context of viral infection versus cancer evolution as HIDDEN cells might themselves evolve. HIDDEN cell effects on basal cells might occur via paracrine activities that create a pro-tumor microenvironment, and particularly in severely dysplastic lesions where ELF3+ HIDDEN cells appear to expand downwards into the basal layer (Fig7F). Additional possible pro-tumor activities are detailed in the response to point 6 of this reviewer, and these are now highlighted in the Discussion (pages 17-18).

While we have obtained and analyzed additional CIN tissues, we were unable to procure CIS lesions or cervical SCC tissues which are exceedingly rare in the clinic.

We have clarified on page 6 in the Results section the difference between transcriptomically distinct clusters and cluster-defining markers. While each scRNAseq cluster is defined cumulatively by a distinct transcriptomic gene list, we sought to identify cluster-defining genes that could serve as biomarkers to independently represent clusters without the need for more labor intensive combinatorial approaches. In the case of C8, there were no genes that were specific to C8, rather a combination of genes from the gene list would be necessary to robustly localize C8. While we recognize this as an important distinction, the language related to C8 biomarkers has been removed as the manuscript is now focused on HIDDEN cells.

The C8 population is of high interest as noted by the reviewer, given that aberrant HPV-induced proliferation has long been described in the literature and is an attractive therapeutic target. We have so far unsuccessfully attempted to define a sufficiently unique signature that distinguishes cycling suprabasal C8 from basal C6 cells. This is in line with the role of the viral E7 oncogene in suprabasal cells which inappropriately activates E2F by mechanisms that closely mimic authentic cell cycle progression in uninfected tissue. C8 cells require additional studies for publication, while a greatly expanded set of rafts, pre-cancer and cancer specimens was added to strengthen focus on C9.

9. *There are misreferenced figures in the text (and the authors should verify all):*

Page 11, columns 1-2: Figs. noted as Fig. 5, should be Fig. 8. Perhaps also on page 13?

Page 11, column 2: "Additionally, expression of C9 markers was increased in models of differentiation (overconfluent 2D cultures and 3D rafts), relative to undifferentiated (subconfluent) 2D cultures (Fig. 1C)." Fig. 1C should be Fig. S1C.

Could not find Fig. S4D and 6 (noted on page 13) or Fig. 6K, 6L (should be 9K and 9L?) noted in the legend to Fig. S1

Fig. S1 seems to use different nomenclature for genes in the legend versus some of the figures (MACC1, CLDN4), which is confusing.

Page 14, column 1 mentions Fig. 7B, but this does not seem to be accurate.

Reference 23 seems to be for reovirus and not HPV.

It would be helpful if the colors of blue for downregulated genes and yellow for upregulated genes in the heat maps were maintained in all cytoscape plots and bar graphs for consistency. (e.g., Fig. 5 C, D seems inverse).

For example, is abbreviated "e.g.", not "eg".

These detailed comments are appreciated, and we have corrected the manuscript correspondingly. The previous Fig5 (now Fig6) was modified to use colors consistent with other figures. We indicate ontology analyses of the entire C9 transcriptome in green (Fig6A-B) and differential analyses within C9 HPV+ vs HPV- cells to match the colors presented in the initial heatmap (Fig 2D), namely yellow for upregulation (Fig. 6C-D) and blue for downregulation. (Fig6E)

Reviewer #2, expertise in scRNA-seq analysis (Remarks to the Author):

Bedard et al. applied scRNA-seq to NIKS (normal immortalized keratinocytes), with and without HPV infection, grown in organotypic 3D rafts, in order to assess the effects of HPV on infected epithelial tissue at single cell resolution. Some of the findings were further validated through RNA-ISH of sections from other NIKS as well as patient samples. The analysis revealed 12 clusters of epithelial cells, with 11 clusters that are shared between HPV+ and HPV- cells and only one that is unique to HPV+ cells. However, multiple clusters are biased towards either HPV+ or HPV- cells. These differences are primarily related to the cell cycle, which is increased, as expected, in HPV+ cells and to differentiation processes. The authors focus on cluster 9 (C9) that is the most distinct from all other clusters, is enriched in (but not specific to) HPV+ cells, has a specific spatial localization to the uppermost layer of HPV-infected rafts, and is characterized by ELF3 mRNA expression. It is then demonstrated that C9 cells are also present in HPV+ cancer tissues, and that ELF3 expression is correlated with prognosis of cervical cancer and that it regulates the C9 cells.

Given the significance of HPV infection and the limited knowledge of HPV effects at the tissue level, the general direction of this study is important and the experimental setup is interesting although not entirely cutting edge.

The experiments appear to be executed at a high level. The analysis is performed at a medium level, which suggests only partial understanding of the methods and suboptimal flow and presentation of the results. The identification of C9 is of interest, although the origin, function and significance of this population remain quite vague. This population is not unique to HPV+ cells, its association with survival does not suggest causation and its biology was only described in a preliminary manner. Overall, the study is of significance but lacks a major finding or conceptual advance.

We thank the Reviewer for their appreciation of the importance of HPV infection and application of single cell transcriptomics to discover HPV-induced reprogramming in heterogeneous keratinocyte populations. The experimental setup is indeed not cutting edge in that we have deliberately chosen the most established NIKS raft model used for HPV biology studies for the scRNAseq experiments. The intent was to capitalize on the proven two decade history of this model to achieve fundamental insights into HPV gene function combined with it being an optimal system for isogenic comparisons in the absence of selective pressure. The benefits and caveats of this model are now discussed in the manuscript as outlined in the response to Reviewer 1 comment 2. We recognize that more robust validation and experimentation was needed to improve the depth and clinical significance of our analysis as related to HPV biology and HPV-driven cancers. To this end, we have extensively quantified and geographically localized the terminally differentiated KLK7+/C3 (Fig. 4) and the HIDDEN/C9 (Fig. 5) subpopulations across three independently generated stratified squamous epithelial rafts. Additionally, we have substantially expanded prior analyses of HPV+ versus HPV- cervical and head and neck tissues through the culture of primary cervical and tonsillar tissues that were then converted into HPV+ and HPV- organotypic rafts to establish the shared presence and localization of the HIDDEN cell subpopulation (Fig 7A-D, Fig. S4A-C, and Fig. 10A-F). We refer R2 to Reviewer 1 comments for additional details and improvements to the studies. We believe that these changes have greatly improved the robustness of the analysis.

In terms of conceptual advances, we believe the following points are major new contributions to the field of HPV infection and disease wherein we 1) report the transcriptomic landscape of high risk HPV infected stratified epithelium for the first time, 2) discover epithelial HIDDEN cells that form a new compartment and have the potential to mark the surface of HPV-infected tissue, 3) identify ELF3 as a required HPV-induced transcriptional driver of the HIDDEN compartment, and 4) detect and associate HIDDEN cell enrichment in HPV+ SCCs with poor prognosis, findings; thus HIDDEN cells may themselves be candidate preventive and/or therapeutic targets. We have included this emphasis in the Discussion (page 18) to help clarify the importance of this work. While elucidation of specific mechanisms and functional roles of HIDDEN cells in cancer pathogenesis awaits further study, our conceptual advancements now form the basis for the easily envisioned potential direct or indirect (via the extracellular environment) pro-tumor activities of this subpopulation discussed in the revised manuscript that will be explored by us and others to translate this work to the clinical arena.

Major comments

1. The manuscript seems long and non-focused with suboptimal flow and too many figures. Accordingly, I found it difficult to read the manuscript and connect the different pieces into a coherent picture. In my opinion, the authors should identify the main and most reliable results that are of broad interest, focus on them while improving the flow and the presentation of those results, and move the remaining analyses to the supplement. A shorter and more focused paper could be considerably improved.

See response to R1, comment 4: "We thank the Reviewer for the suggestion to consider separating the manuscript into two separate manuscripts given the expansiveness of the data and lengthy results section. Thus, we have focused this manuscript on the HIDDEN and KLK7+ subpopulations enabling a more concise, understandable results section complemented by an expanded Discussion of the major findings together with the biologic and clinical relevance. To this end, we have extensively revised the results section with three entirely new main figures (Fig. 7, 9, 10), and added many new sub-figures to previous figures to include KLK7+/C3 validation (Fig4F-G, previously Fig3) and HIDDEN/C9 validation (Fig5D,E,F,H,I and Fig S2D,E; previously Fig6 and FigS4). The previous Figures 4,5, 7 and 8 and supplementary figures Fig S2 and Fig S3 were removed for a separate manuscript as recommended by Reviewers 1 and 2 to focus the manuscript and enhance the understanding of the primary discoveries by a broader audience."

2. Pseudotime analysis should be used and interpreted with caution and it is not suitable for connecting well-

separated clusters that do not lie on a continuous trajectory. In this particular case, C9 seems completely distinct from the other clusters and is “forced” onto the trajectory. Its inclusion in the pseudotime analysis is problematic and the results are not reliable.

We appreciate this comment, and completely agree. We have now included pseudotime analysis wherein C9 is removed prior to ordering the cells along pseudotime (Fig. 3A-D). In this analysis, all cells can more clearly be shown to fall into a linear differentiation progression. We have retained the previous analysis in Fig S1 in order to reinforce that C9 is completely distinct from the trajectory and cannot be analyzed using pseudotime. We have additionally emphasized this point on page 5.

3. To demonstrate the relevance of C9 for in vivo HPV+ disease the authors analyzed normal and disease tissues by IF and ISH with markers of C9. However, the IF results cannot be directly compared to the ISH results. Such comparative analyses are only relevant if they rely on exactly the same protocol.

We performed RNA-ISH and IF variably across different tissue types and did not intend to claim that there is complete overlap between mRNA and protein expression. We recognize that IF and ISH were not consistently used across the different tissue types. We have now performed both RNA-ISH and IF on the new cervical (Fig. 7A-D) and tonsillar rafts (Fig. 10A-F, Fig. S4A-C) as well as added IF data to supplement previously shown RNA-ISH data on head and neck tissues (Fig 10G-H). The revised manuscript now includes both ISH and IF results for both cervical and head and neck tissues with direct negative controls.

4. The authors argue that the association of C9 with survival cannot be analyzed in HNSCC due to variation between sites and limited sample numbers. The first issue can be avoided by analyzing each site separately and focusing on the sites with the highest statistical power. The second issue can be approached by presenting the results despite their limited statistical power. Even if the effect is not significant I would argue that the data should be shown and interpreted in light of the limited statistics. For example, a non-significant trend towards C9 correlating with poor survival would still be partially consistent with such association, while a complete lack of correlation or a non-significant opposite trend (towards the correlation in the opposite direction) may not be consistent with such association even with limited statistical power.

In response to this suggestion, we selected the ICD-10 subsite with the most HPV+ HNSCC cases (C09.9, tonsil, 32 cases). Of these, 7 harbored high mRNA expression of the co-occurring biomarkers and 25 harbored unaltered expression. Unfortunately these cases had a high frequency of censoring, which refers to incomplete survival data in the TCGA (5 of the 7 high mRNA cases, and 19 of the 25 unaltered cases). This left only 2 cases of high biomarker expression to compare to 6 cases of unaltered mRNA expression. Survival was not found to be statistically different for the 2 cases, but we feel that the exceedingly low number of patients precludes a meaningful conclusion. Similarly, in the subsite with the next highest number of HPV+ HNSCC cases (C01, base of tongue, 11 cases) there were 3 cases with high mRNA expression and 8 with unaltered expression. However, all 11 cases are censored and survival studies at this subsite are therefore not possible. The remaining subsites with 3 or more HPV+ cases are C02.9 (3/6 cases censored), C10.9 (4/5 cases censored), C14.8 (3/5 cases censored), and C32.9 (3/3 cases censored). Based on these numbers, we do not feel that analyses of separate sites will be informative for the readers.

5. As I understand the experimental setup, HPV-infected and non-infected NIKS were sequenced together and the NIK of origin was determined by HPV16 mRNA expression. While the authors do show ubiquitous HPV16 DNA expression this does not necessarily translate to HPV reads being identifiable when performing scRNAseq, especially with the known problem of dropouts. Thus, when the authors compare HPV+ and HPV- cells between different clusters, some apparent HPV- cells could actually originate from the HPV-infected NIK, and some of the clustering could be due to batch effects.

The experimental setup was insufficiently explained in the manuscript, and this has been corrected. HPV-infected and non-infected NIKS were sequenced separately but analyzed jointly such that the origin of each cell from either HPV negative or positive tissue is known (clarified on pages 4 and 25).

6. Did any of the HPV-infected cells undergo further carcinogenic transformation? This may be explored by pathology analysis, inferring copy number aberrations etc.

HPV16-infected and uninfected NIKS were cultured on irradiated feeder cells to provide optimal growth conditions and to prevent viral integration which is a hallmark of carcinogenesis. These cells are not expected to routinely undergo transformation and have not been reported in the literature to transform.

7. In figure 5A ribosomal genes are shown to be enriched in HPV- cells. In scRNAseq, ribosomal genes often indicate a quality issue rather than actual biology: since they are the most highly expressed genes in cells, they are often affected by the global quality and coverage of single cell data and may appear as upregulated or downregulated due to technical reasons. The fact that figure 8H shows mitochondrial genes, a proxy for dying cells, to be upregulated in HPV- strengthens this point, as does the generous QC threshold of 30% mitochondrial genes per cell. The authors should show that QC parameters are comparable between HPV- and HPV+ cells and ensure that the results are robust to the use of more strict QC filtering.

We thank the reviewer for pointing this out. In fact, our QC threshold was mistakenly reported as 30% mitochondrial genes per cell. We have reviewed the data to verify that the threshold was instead 20% and have corrected this in the corresponding Methods on page 25.

Minor comments

1. It may be helpful to name the clusters with informative names based on their top marker genes.

We appreciate the reviewer's suggestion to improve the nomenclature for the epithelial subpopulations of interest. As the manuscript now features a refined focus on C3 and C9, we refer to C3 as the most terminally differentiated KLK7+ epithelial cells (page 6) and to C9 as HIDDEN (HPV-induced dissociation-dissonant epithelial non-conventional) cells (page 10) and believe that this greatly improves and clarifies Discussion of these subpopulations.

2. Functional studies of the impact of ELF3 overexpression/depletion on tissue phenotype are not necessary but could improve the study.

We regret that we were unable to complete these suggested experiments to further add to the study within the confines of the time frame provided for re-submission.

3. In figure 2F, HPV+ cells are said to be enriched in certain clusters. This should be done stringently with a statistical test and p-value shown

The reviewer points out that quantification of the enrichment of HIDDEN cells in HPV infected tissues vs non-infected counterparts is an important validation that requires stringent statistical tests and p-value. We were initially interested in C9/HIDDEN since the percentage of cells with the HIDDEN cell transcriptome per raft was enriched in HPV+ (7.0%) versus HPV- (3.1%) epithelium in the scRNAseq data, at a 126% increase. However, given that these numbers represent one sample each and that cell numbers per cluster from scRNAseq data are not necessarily representative of the composition of the dissociated sample, we focused on quantifying compartment size and geographic location in 3 independently derived rafts directly via RNA-ISH and IF (new Fig 4F-G for KLK7+ C3 cells, and new Fig5D-F and H-I for HIDDEN/C9 cells). These experiments validated our observations for HIDDEN and KLK7+ cells from the RNAseq data with statistical tests and p values that are now described in the Materials and Methods (page 28) and throughout the relevant figure legends.

4. Figure 4B and other similar figures show two repeats with no significance values. More repeats and statistical testing should be added.

We agree that more replicates and significance values would strengthen the paper. KLK7+/C3 and HIDDEN/C9 cells were detected by RNA-ISH in 3 biologically independent experiments and we have now carefully quantified this new data for total signal intensity and geography in figures 4 and 5.

5. P-values and statistical tests used for GO enrichment analysis should be shown.

The GO enrichment analysis statistical analysis was insufficiently described in the manuscript. The gProfiler web server was used to perform enrichment analysis. We have included a brief description in the ontology analysis Methods section describing the statistical methods used to calculate the P-values for Data S4 (page 26).

6. If no C8-specific markers were found, what makes this a distinct population?

See response to R1, point 8: “We have clarified on page 6 in the Results section the difference between transcriptomically distinct clusters and cluster-defining markers. While each scRNAseq cluster is defined cumulatively by a distinct transcriptomic gene list, we sought to identify cluster-defining genes that could serve as biomarkers to independently represent clusters without the need for more labor intensive combinatorial approaches. In the case of C8, there were no genes that were specific to C8, rather a combination of genes from the gene list would be necessary to robustly localize C8. While we recognize this as an important distinction, the language related to C8 biomarkers has been removed as the manuscript is now focused on HIDDEN cells.”

The C8 population is of high interest as noted by the reviewer, given that aberrant HPV-induced proliferation has long been described in the literature and is an attractive therapeutic target. We have so far unsuccessfully attempted to define a sufficiently unique signature that distinguishes cycling suprabasal C8 from basal C6 cells. This is in line with the role of the viral E7 oncogene in suprabasal cells which inappropriately activates E2F by mechanisms that closely mimic authentic cell cycle progression in uninfected tissue. C8 cells require additional studies for publication, while a greatly expanded set of pre-cancer and cancer specimens was added to strengthen focus on C9.

7. On page 11, references to figure 5 actually refer to figure 8.

We apologize for the oversight and have corrected this error.

Reviewer #3, expertise in 3D organotypic epithelial raft models (Remarks to the Author):

In this manuscript, the authors aimed to explore the keratinocyte subpopulations which support distinct phases of the viral life in persistent HPV16 infection. To this purpose, the authors generated a HPV16-isogenic organotypic epithelial raft model as it mimics the effects of HPV infection on epithelial biology. By scRNAseq to generate a host pathogen transcriptome atlas of HPV16+ versus HPV16- keratinocytes, an HPV-reprogrammed keratinocyte subpopulation (C9) has been identified in a new surface compartment of the squamous epithelium. The author found that this subpopulation required overexpression of the ELF3/ESE-1 transcription. Moreover, TCGA analyses of patient tissues demonstrated that high ELF3 expression levels were associated with HPV in premalignant and malignant disease. The final ELF3 knockdown experiments suggest that ELF3 is required for formation of the C9 compartment at the surface of stratified HPV16+ epithelium. The authors conclude that ELF3 might be a potential therapeutic target in the fight against HPV persistence and carcinogenesis.

The combined use of HPV-infected 3D culture technique and scRNAseq is very interesting. In my opinion the association between the new single cell technology and the biological approach has generated a significant biological insight into the cell populations under study for such a high-impact general interest journal. The analyses and methods performed are complete and consistent with answering the specific question. The results are robust and fit-for-purpose and the controls and sampling mechanisms are sufficient and well described.

I recommend this manuscript for publication with no revisions.

We thank the reviewer for their enthusiasm for this manuscript.

Reviewers' Comments:

Reviewer #1:

Remarks to the Author:

Dear Editor,

I have carefully reviewed the response to the reviewers and the revised manuscript by Bedard et al. This is a novel and informative study that will be of broad interest to the HPV field, as well as to those studying epithelial biology and epithelial cancers.

The authors have thoughtfully responded to the prior critiques and the manuscript is vastly improved in focus and content. The addition of HPV18 cervical and HPV16 tonsillar rafts, CIN, and HNSCC tissues significantly enhances the rigor and impact of this study.

I did not see information on the probe used for HPV DNA ISH; presumably the noncoding LCR, but this should be noted.

There is no indication that tissues were DNase treated prior to RNA-ISH. The authors should note that it is very difficult to ensure complete DNase removal of viral genomes from highly differentiated cells, presumably because genomes may be partially protected from DNase by viral capsids. Thus, we have had difficulty quantitatively removing viral genomes from raft tissues. This is problematic because all RNA ISH probes will detect genomes in the leaky nuclei of upper cells. This caveat should be mentioned, and the authors should comment on whether they can be certain that the L1 and E7 ISH signals are cytoplasmic, which would be expected for RNA specificity. This is more important for interpreting whether E7 RNAs are in the upper cells since capsid protection of viral genomes would be consistent with the presence of L1 RNAs.

There are some instances of inconsistent font usage or word spacing that should be corrected (in case the journal editorial office does not provide this service).

"clusters" page 5 column 2.

"rafts" page 5, column 1, first paragraph

Not fully justified on page 12, column 1.

Page 14, second paragraph would benefit from directing the reader to specific figure panels.

Page 17 "Such activities may act on aspiring tumor cells to promote proliferation or survival, or may reprogram the tumor microenvironment to promote angiogenesis or..." would "aspiring malignant cells" be more appropriate since the productive HPV infection is functionally a benign tumor?

Reviewer #2:

Remarks to the Author:

The revised version of the manuscript by Bedard et al. is significantly improved. It is more focused, emphasizing the C9 population as the central result, and includes new experiments and analyses that further strengthen the relevance and significance of the C9 population, such as demonstrating the presence of similar populations of cells in HNSCC. I think that these improvements make the manuscript more suitable for publication, but I still have several comments, mostly relating to new analysis and the significance of the C9 population.

Specific comments:

1. The flow of the revised version is improved and is more focused. Yet, especially with all of the

additions, it remains a long and detailed manuscript that I therefore find somewhat difficult to read. I would suggest the authors to move many of the results to the supplement and to ensure that the text is more concise and focused on the most important parts.

2. The demonstration that cells similar to those of C9 are found in HNSCC is an important addition that demonstrates the relevance of C9 to disease and can go deeper than other analysis (e.g. TCGA) given the availability of single cell data for those cells. Yet, the current analysis seems somewhat preliminary and should be significantly extended.

First, the authors use batch correction to combine the tumors, but tumors from different patients are biologically distinct in a meaningful way and such inter-patient variability is effectively removed by the batch correction. Thus, the authors should repeat the analysis of the cancer cells without batch correction and show the results with both approaches.

Second, all cancer cells are pooled together and it is never shown which cells come from which tumor and how tumor-specificity is linked to the cells similar to C9. Specifically, are cells similar to C9 found across all/most tumors (in both approaches - with and without batch correction) or could such cells be unique to a small subset of tumors? As a related side note, the authors should not refer to the cells in the tumor as "keratinocytes" as these are transformed cells that only resemble keratinocytes but should be named more accurately.

Third, the current analysis only shows that some cells score highly for the C9-based signature but does not use those HNSCC cells to re-define the analogous signature to C9. In other words, it would be important to define the signature of these cells in HNSCC and then to compare the signature to the author's model and examine similarities and differences. To this end, it would be helpful to add heatmaps (with all the genes of the signature) to compare individual cancer cells with high and those with low expression of the C9 signature taken from the same tumors.

Fourth, if a population similar to C9 indeed exists in published HNSCC datasets, then was that population described previously by the original study of the data used for this analysis? or by other single cell studies of HNSCC (I believe that a few such studies were already published)? if not, then can the authors explain why was that population not identified by the previous studies? If these cells are indeed so unique within HNSCC, then it would be difficult for me to understand why they were not described previously; and if they were described, then I believe that it is important to acknowledge that these cells have been reported previously and thereby improve the convergence between this and previous studies. Indeed, one of the main problems in the single cell field is that each study gives their own names and classifications for the same cell types and cellular states that are seen by other studies, resulting in a lot of confusion and in difficulty to resolve the specific studies and generate a robust perspective.

3. Association of C9 cells with poor prognosis: this is highlighted as a central result, but is only supported by bulk TCGA cervical data. If the association was also observed in HNSCC and/or was demonstrated by single cell data then I would agree that this is a strong and important result. But since it is only seen in cervical cancer and even in that context it is only demonstrated with analysis of bulk samples (where it is not clear if the result directly reflects a subpopulation similar to C9 or alternatively a higher overall expression of some of those genes across the entire tumor), then I would argue that this result should be described as preliminary supporting evidence and discussed more carefully.

4. In my opinion, both the title of the manuscript and the abbreviation of the C9 population are suboptimal. In the title "A single cell transcriptome atlas...", it seems less appropriate to me to describe this work as an atlas, which is usually reserved to describe systematic datasets with a large number of samples. I would suggest to rephrase as "Single cell analysis of ...".

The C9 population is called HPV-induced differentiation dissonant epithelial nonconventional (HIDDEN) cells. This is a long name that is difficult for me to understand and cannot be easily remembered. It also seems somewhat misleading to name those cells as "HPV-induced" when they also exist among HPV- cells. and finally, it seems like this name was selected in order to fit the HIDDEN abbreviation, which could also be misleading because those cells are not hidden in any way beyond other populations of cells. I would therefore suggest to rename them, for example as Abnormal Differentiation Keratinocytes (ADK) or something similar to that. I would also suggest the authors to be careful with statements that could be misinterpreted as implying that these cells are specific to HPV+ cells. For example, the line in page 3 " ...discovering a novel HPV-reprogrammed subpopulation" implies to a naive reader that these cells are generated by HPV-based reprogramming and would not exist in HPV- populations.

Response to reviewer comments. We thank the reviewers for their comments on the revised version of the manuscript, and have provided responses to each below. Corresponding changes in the revised manuscript are indicated in green, while previous changes remain in blue. Reviewer 1 considered the manuscript novel, informative and of broad interest, and appreciated our responses to previous critiques. We have addressed remaining technical questions regarding DNAish and RNA-ISH, and have corrected inconsistent fonts and spacing in the text. We addressed and responded to additional questions by Reviewer 2 who also appreciated the revised manuscript for the significant improvements.

Reviewer #1 (Remarks to the Author):

1. I have carefully reviewed the response to the reviewers and the revised manuscript by Bedard et al. This is a novel and informative study that will be of broad interest to the HPV field, as well as to those studying epithelial biology and epithelial cancers.

The authors have thoughtfully responded to the prior critiques and the manuscript is vastly improved in focus and content. The addition of HPV18 cervical and HPV16 tonsillar rafts, CIN, and HNSCC tissues significantly enhances the rigor and impact of this study.

We thank the reviewer for their enthusiasm for the revised version of the manuscript.

2. I did not see information on the probe used for HPV DNA ISH; presumably the noncoding LCR, but this should be noted.

The probe used for HPV DNA-ISH is now described on page 26 in the methods section.

3. There is no indication that tissues were DNase treated prior to RNA-ISH. The authors should note that it is very difficult to ensure complete DNase removal of viral genomes from highly differentiated cells, presumably because genomes may be partially protected from DNase by viral capsids. Thus, we have had difficulty quantitatively removing viral genomes from raft tissues. This is problematic because all RNA ISH probes will detect genomes in the leaky nuclei of upper cells. This caveat should be mentioned, and the authors should comment on whether they can be certain that the L1 and E7 ISH signals are cytoplasmic, which would be expected for RNA specificity. This is more important for interpreting whether E7 RNAs are in the upper cells since capsid protection of viral genomes would be consistent with the presence of L1 RNAs.

This is an important point - the standard RNAscope protocol was used which does not include a DNase step. We have carefully assessed the image in Fig1D, and concluded that we cannot be certain that L1 and E7 signals are cytoplasmic. Because the Southern blot analysis, DNAish, and RT-qPCR all support a nonproductive life cycle as validated subsequently by the scRNAseq data, and because the RNA-ISH did not add additional information, we have removed subfigure D from Figure 1. We appreciate this comment.

There are some instances of inconsistent font usage or word spacing that should be corrected (in case the journal editorial office does not provide this service).

- *“clusters” page 5 column 2.*
- “rafts” page 5, column 1, first paragraph*
Not fully justified on page 12, column 1.

Page 14, second paragraph would benefit from directing the reader to specific figure panels.

Page 17 “Such activities may act on aspiring tumor cells to promote proliferation or survival, or may reprogram the tumor microenvironment to promote angiogenesis or...” would “aspiring malignant cells” be more appropriate since the productive HPV infection is functionally a benign tumor?

We have made all of the suggested changes.

Reviewer #2 (Remarks to the Author):

The revised version of the manuscript by Bedard et al. is significantly improved. It is more focused, emphasizing the C9 population as the central result, and includes new experiments and analyses that further strengthen the relevance and significance of the C9 population, such as demonstrating the presence of similar populations of cells in HNSCC. I think that these improvements make the manuscript more suitable for publication, but I still have several comments, mostly relating to new analysis and the significance of the C9 population.

We appreciate the Reviewer's enthusiasm for the extensive re-organization and changes in the revised manuscript.

Specific comments:

4. The flow of the revised version is improved and is more focused. Yet, especially with all of the additions, it remains a long and detailed manuscript that I therefore find somewhat difficult to read. I would suggest the authors to move many of the results to the supplement and to ensure that the text is more concise and focused on the most important parts.

We have moved Table 1 to Supplemental Data to assuage this concern. We feel that the remaining figures are key to understanding the manuscript, and that relegating these to the supplement would make it more and not less difficult to comprehend. However, we will seek editorial advice on this issue.

5. The demonstration that cells similar to those of C9 are found in HNSCC is an important addition that demonstrates the relevance of C9 to disease and can go deeper than other analysis (e.g. TCGA) given the availability of single cell data for those cells. Yet, the current analysis seems somewhat preliminary and should be significantly extended.

First, the authors use batch correction to combine the tumors, but tumors from different patients are biologically distinct in a meaningful way and such inter-patient variability is effectively removed by the batch correction. Thus, the authors should repeat the analysis of the cancer cells without batch correction and show the results with both approaches.

This analysis was done as recommended, and the corresponding data is added as Fig S5A, with details described on page 13. In the absence of batch correction, as published in the original article by the Ferris group (PMID: 34921143), HNSCC epithelial cells cluster primarily by tumor specimen.

Second, all cancer cells are pooled together and it is never shown which cells come from which tumor and how tumor-specificity is linked to the cells similar to C9. Specifically, are cells similar to C9 found across all/most tumors (in both approaches - with and without batch correction) or could such cells be unique to a small subset of tumors? As a related side note, the authors should not refer to the cells in the tumor as "keratinocytes" as these are transformed cells that only resemble keratinocytes but should be named more accurately.

This is an important point. We performed additional analyses as requested, and have added the new Fig 9F to demonstrate that all but one tumor (HPV- tumor HN5) contribute to the C7 subpopulation in the Ferris dataset that contains the strongest expression of the C9 signature. We also find that C7 was composed of a greater proportion of cells derived from HPV+ vs HPV- specimens, consistent with upregulation of the C9 signature in HPV+ tumors. We also now refer to the cells in the tumor as epithelial cells.

Third, the current analysis only shows that some cells score highly for the C9-based signature but does not use those HNSCC cells to re-define the analogous signature to C9. In other words, it would be important to define the signature of these cells in HNSCC and then to compare the signature to the author's model and examine similarities and differences. To this end, it would be helpful to add heatmaps (with all the genes of the signature) to compare individual cancer cells with high and those with low expression of the C9 signature taken from the same tumors.

Our very preliminary data suggest that, as might be expected, epithelial tumor cells with a C9/HIDDEN cell signature have evolved when compared to their corresponding counterparts in infected squamous epithelium. How this evolution produces transcriptional (and associated functional) consequences will be the exciting subject of a new manuscript. We believe that the addition of new analyses and heatmaps will open up new research questions and complexities, while still not addressing the significance of C9. Furthermore, we find ourselves in a quandary between now having added 39 new subfigures based on reviewer suggestions, and Reviewer 2's valid point that the manuscript is already data dense (see point #4).

Fourth, if a population similar to C9 indeed exists in published HNSCC datasets, then was that population described previously by the original study of the data used for this analysis? or by other single cell studies of HNSCC (I believe that a few such studies were already published)? if not, then can the authors explain why was that population not identified by the previous studies? If these cells are indeed so unique within HNSCC, then it would be difficult for me to understand why they were not described previously; and if they were described, then I believe that it is important to acknowledge that these cells have been reported previously and thereby improve the convergence between this and previous studies. Indeed, one of the main problems in the single cell field is that each study gives their own names and classifications for the same cell types and cellular states that are seen by other studies, resulting in a lot of confusion and in difficulty to resolve the specific studies and generate a robust perspective.

The C9 population was not previously described by any single cell study. Overall, there is a paucity of reports that specifically detect epithelial cells in HPV+ vs HPV- HNSCCs, which would have identified the C9/HIDDEN cell signature. Our literature review finds that many publications are limited in this regard by (1) exclusion of epithelial cells for scRNAseq capture, (2) missing HPV status, (3) overwhelming focus on immune processes. First, several studies such as Wieland *et al.* (PMID: 33208941) and Cillo *et al.* (PMID: 31924475), isolate immune cells by sorting prior to scRNAseq analysis and thus do not capture any data on epithelial cells. Second, of those studies that include epithelial cells, not all utilize HPV-typed specimens and some focus on anatomical sites wherein HPV-positive cases are rare, eg. in the oral cavity. Third, to our knowledge, the only study which captures all cell types in a panel of HPV+ and HPV- HNSCCs is by Kürten *et al* (PMID: 34921143). However, this report was exclusively focused on diverse immune cell populations and did not analyze epithelial cells. We therefore selected this dataset for our analyses focused on epithelial HNSCC cells harboring a C9 signature. Finally, many other reports simply re-analyze available GEO and TCGA data and thus suffer from the same limitations outlined above. We appreciate this comment as it further highlights the importance of our analyses, and this is now discussed on page 17 in the Discussion.

6. Association of C9 cells with poor prognosis: this is highlighted as a central result, but is only supported by bulk TCGA cervical data. If the association was also observed in HNSCC and/or was demonstrated by single cell data then I would agree that this is a strong and important result. but since it is only seen in cervical cancer and even in that context it is only demonstrated with analysis of bulk samples (where it is not clear if the result directly reflects a subpopulation similar to C9 or alternatively a higher overall expression of some of those genes across the entire tumor), then I would argue that this result should be described as preliminary supporting evidence and discussed more carefully.

This is an excellent point. We have now added a discussion of associated caveats and mentioned the preliminary nature of this analysis in the Discussion.

7. *In my opinion, both the title of the manuscript and the abbreviation of the C9 population are suboptimal. In the title "A single cell transcriptome atlas...", it seems less appropriate to me to describe this work as an atlas, which is usually reserved to describe systematic datasets with a large number of samples. I would suggest to rephrase as "Single cell analysis of ...".*

This was indeed an oversight. In removing analyses of other cell populations, we neglected to adjust the title. The new title is: "Single cell transcriptomic analysis of HPV16-infected epithelium identifies a keratinocyte subpopulation implicated in cancer." We have correspondingly revised the text of the summary (page 2), introduction (page 3) and discussion (page 16).

8. *The C9 population is called HPV-induced differentiation dissonant epithelial nonconventional (HIDDEN) cells. This is a long name that is difficult for me to understand and cannot be easily remembered. It also seems somewhat misleading to name those cells as "HPV-induced" when they also exist among HPV- cells. and finally, it seems like this name was selected in order to fit the HIDDEN abbreviation, which could also be misleading because those cells are not hidden in any way beyond other populations of cells. I would therefore suggest to rename them, for example as Abnormal Differentiation Keratinocytes (ADK) or something similar to that. I would also suggest the authors to be careful with statements that could be misinterpreted as implying that these cells are specific to HPV+ cells. For example, the line in page 3 " ...discovering a novel HPV-reprogrammed subpopulation" implies to a naive reader that these cells are generated by HPV-based reprogramming and would not exist in HPV-populations.*

We do want to be exceedingly careful throughout the manuscript to point out that C9 cell numbers are greatly induced/increased by HPV beyond a very low baseline, while the C9 compartment, historically equated to the uninfected terminally differentiated (C3) compartment, is novel and unique to HPV infection. The statement on page 3 was misleading, and we clarify that C9 cells are a keratinocyte subpopulation that is greatly amplified and also transcriptionally reprogrammed (Fig9) by HPV. We have scanned the manuscript to avoid similarly confusing language. However, neither C9 cells nor their compartment have ever been described, and it is our opinion that the abbreviation captures this fact, as well as the nature of the cells optimally, and is easily remembered by the readers.

Reviewers' Comments:

Reviewer #2:

Remarks to the Author:

I am satisfied with the authors' responses and have no additional comments.